# Heritability and interindividual variability of regional structure-function coupling

Zijin Gu [1], Keith Wakefield Jamison[2], Mert Rory Sabuncu[1,2] & Amy Kuceyeski [2✉]

White matter structural connections are likely to support flow of functional activation or functional connectivity. While the relationship between structural and functional connectivity profiles, here called SC-FC coupling, has been studied on a whole-brain, global level, few studies have investigated this relationship at a regional scale. Here we quantify regional SC-FC coupling in healthy young adults using diffusion-weighted MRI and resting-state functional MRI data from the Human Connectome Project and study how SC-FC coupling may be heritable and varies between individuals. We show that regional SC-FC coupling strength varies widely across brain regions, but was strongest in highly structurally connected visual and subcortical areas. We also show interindividual regional differences based on age, sex and composite cognitive scores, and that SC-FC coupling was highly heritable within certain networks. These results suggest regional structure-function coupling is an idiosyncratic feature of brain organisation that may be influenced by genetic factors.

[1] School of Electrical and Computer Engineering, Cornell University, Ithaca, NY, USA. [2] Department of Radiology, Weill Cornell Medicine, New York, NY, USA. ✉email: amk2012@med.cornell.edu

The question of how anatomy and physiology are related is one of the fundamental questions in biology, particularly in neuroscience where studies of form and function have led to fundamental discoveries. In the last few decades, the invention of MRI has enabled in vivo investigation of whole-brain, anatomical (white matter) and physiological (functional co-activation) brain networks in human populations. Studies analysing multi-modal connectivity networks have produced a consensus that, to some extent, alignments exist between the brain's anatomical structural connectome (SC) and its physiological functional connectome (FC)[1–5]. Recent work has focused on implementing computational models, including neural mass models, network diffusion models, graph theoretical or statistical approaches, that formalize the global relationship between SC and FC in both healthy and disordered populations[6–9]. Some of the main goals in joint structure–function connectome modeling are to understand how neural populations communicate via the SC backbone[7], how functional activation spreads through the structural connectome[8], to increase the accuracy of noisy connectivity measurements, to identify function-specific subnetworks[10], to predict one modality from the other[1] or to identify multi-modal mechanisms of recovery after injury[11,12]. While useful, these modeling approaches are global in nature and ignore the regional variability in the structure–function relationship that, to date, has not been adequately quantified in adult populations.

Recent publications mapping connectome properties to cognitive abilities have focused on using either FC or SC alone, or concatenating both together to reveal brain-behavior relationships[13–17]. Some recent studies have identified relationships between global, whole-brain SC-FC correlations and cognitive abilities or states of awareness. One such paper showed that stronger global SC-FC correlations were related to worse cognitive function in older adults with cognitive impairment[18]. Another study showed disorders of consciousness patients with fewer signs of consciousness had longer dwell times in dynamic FC states that were most similar to SC[19]. It has also been shown that SC-FC similarity decreases with increasing awareness levels in anesthetized monkeys[20] and, similarly, decreases from deep sleep to wakefulness in humans[21]. Two studies, in severe brain injury and mild traumatic brain injury, revealed that increasing "distance" between SC and FC was related to better recovery after injury[11,12]. These studies all suggest a weaker coupling of SC and FC is related to better cognitive performance and increasing awareness/consciousness. In contrast, however, a recent study showed increased cognitive flexibility was associated with increased alignment of FC and SC[22]. Therefore, how SC–FC coupling relates to various cognitive functions, awareness or other brain states may vary with the behavioral measure and population in question.

Even fewer studies have explored how the strength of the relationship between SC and FC may vary with age and sex. One such study in a small number of subjects ($N = 14$, 18 months to 18 years of age) showed increasing age was strongly related to higher global correlations between SC and FC ($r = 0.74$, $p < 0.05$)[23]. In one of the few studies to date of regional SC–FC coupling, Baum et al.[24] studied a large number of developing subjects ($N = 727$, aged $8-23$ years old) and showed that the relationship between age and SC–FC coupling varied across brain regions, with some regions showing positive and fewer regions showing negative relationships. Furthermore, they showed that stronger SC–FC coupling in rostro-lateral prefrontal cortex specifically was associated with development-related increases in executive function. Another of regional SC–FC coupling analyzed data from a group of around 100 young adults and showed that, overall, regional SC–FC coupling was stronger in females than in

males and that there were sex-specific correlations of SC–FC coupling with cognitive scores[25].

Several recent publications have revealed the varying degrees to which the brain's FC[26–28] and white-matter microstructure, measured with diffusion MRI summary statistics like fractional anisotropy and mean diffusivity, are heritable[29,30]. Very few studies explore heritability of SC networks; however, some recent preliminary work investigated the relationships between gene co-expression, single nucleitide polymorphisms (SNPs), FC, and SC in a developmental cohort[31]. In particular, this recent work suggests that gene co-expression and SNPs are consistently more strongly related to FC than to SC, and furthermore, that the brain's FC architecture is potentially the mediating factor between genetic variance and cognitive variance across the developing population. However, none of these studies have investigated the heritability of regional SC–FC coupling.

These studies of global, whole-brain SC-FC correlations, while informative, largely ignore regional variability of SC–FC coupling that may provide a more detailed picture of how anatomy and physiology vary with age, sex, genetics and cognitive abilities. There are only two studies to date investigating regional SC–FC coupling. The first used task-based FC in an adolescent population, focused on the cortex and did not assess heritability or sex differences[24] while the second used a data from a moderately sized sample of young adults, did not consider the cerebellum and did not investigate the heritability of SC–FC coupling[25].

In this work, we quantify the cortical, subcortical and cerebellar topography of SC–FC coupling at rest in a group of young adults, verify its reproducibility and quantify its association with age, sex and cognition. Moreover, due to the nature of the HCP data, we were also able to assess the patterns of heritability of regional SC–FC coupling using kinship data. Accurate quantification of the relationship between the brain's structural and functional networks at a regional level is imperative so we can understand how interacting brain circuits give rise to cognition and behavior, and how these relationships can vary with age, sex, cognition and genetics.

## Results

We begin by presenting the regional SC–FC coupling values across unrelated young adults, comparing whole-brain SC–FC coupling to between- and within-network SC–FC coupling, and demonstrating this measure's within-subject and out-of-sample reliability. We then map the regional relationships between whole-brain SC–FC coupling and age, sex and cognition. Finally, we demonstrate the heritability of whole-brain SC–FC coupling. Our data is comprised of MRI, demographic, cognitive and familial relationship data from a group of 941 young and healthy adults, curated by the Human Connectome Project[32] (HCP). Individuals from the HCP's S1200 release were included if they had four functional MRI scans, a diffusion MRI scan and a Total Cognition test score, see Supplementary Fig. 1 for details. A fine-grained atlas (CC400)[33] was used to partition the brain into 392 spatially contiguous, functionally defined cortical and sub-cortical regions. Two 392 × 392 weighted adjacency matrices were then constructed, representing whole-brain SC and FC. FC was calculated via Pearson correlation of the regional time series. SC matrices were constructed using anatomically constrained probabilistic tractography; entries in the SC matrices were then a sum of the global filtering weights (SIFT2) of streamlines connecting pairs of regions, divided by the sum of the volumes of the two regions. Once the FC and SC were constructed, the regional SC–FC coupling vector was calculated for each individual in the following way. Each row in the SC matrix, representing a region's

SC to the rest of the brain, was correlated (via Spearman-rank) with the same region's row in the FC, providing a regional SC–FC coupling vector of length 392 for each subject (Fig. 1). We chose to use Spearman-rank correlation as it is straightforward to interpret, non-parametric (entries in SC are not Gaussian), and, furthermore, enables direct comparison of our results to previous work[24,25]. This whole-brain measure of SC–FC coupling reflects the alignment of a region's functional and structural connectivity profiles to every other region in the brain, but it does not disentangle the contribution of between or within-network connections to the whole-brain coupling value. To assess the association between whole-brain SC–FC coupling and between and within-network coupling, we separately calculated, for each region, its between and within-network SC–FC coupling. Within-network SC–FC coupling for each region was the Spearman correlation of the structural and functional connections between that region and other regions in the same network; between-network SC–FC coupling the same calculation but between that region and regions outside of it's assigned network.

**SC–FC coupling varies spatially, is consistent over time and is reproducible.** The group average SC–FC coupling over 420 unrelated individuals is shown in Fig. 2a. We found that, at the group level, regional SC–FC coupling was almost entirely positive and varied greatly across cortical and subcortical areas, ranging from −0.01 to 0.42. Visual and subcortical areas generally had higher SC–FC coupling than the other networks (see Fig. 2b, c), with values of $0.24 \pm 0.07$ and $0.24 \pm 0.08$, while limbic and default mode areas had significantly weaker SC–FC coupling than the other networks (see Fig. 2b, c, all FDR corrected $p < 0.05$), with values of $0.11 \pm 0.04$ and $0.14 \pm 0.08$. When comparing whole-brain SC–FC coupling to the within and between-network coupling, we found that, unsurprisingly, whole-brain coupling was highly correlated with the between-network SC–FC coupling

(Pearson's $r = 0.704$, $p = 0$) and moderately correlated with the within-network coupling (Pearson's $r = 0.416$, $p = 0$), see Supplementary Fig. 2. This is likely due to the much larger number of between-network region-pairs than within-network region-pairs in the whole-brain SC–FC coupling calculations. Finally, we observed that SC–FC coupling was also moderately positively correlated with SC node degree (Pearson's $r = 0.281$, $p = 0$) but not correlated with FC node degree (see Supplementary Fig. 3).

Next, we tested the reliability and reproducibility of SC–FC coupling by examining its consistency within individuals over time and across different populations of individuals. To test for consistency over time within the same individuals, we used data from a subset of 41 subjects who had a second MRI 6 months after the first. SC–FC coupling was indeed highly consistent across this time period, with a mean difference of $\mu = -0.002$, limits of agreement LoA $= \mu \pm 0.034$, see Fig. 3a, and a test–retest correlation of 0.977 ($p = 1.397e - 264$). Furthermore, we examined out-of-sample, across population reliability in SC–FC coupling using a subset of 346 unrelated HCP subjects (age, $28.78 \pm 3.80$ y; 148 males and 198 females), distinct from the initial set of 420 unrelated subjects. Out-of-sample reliability was also high, with a small mean difference $\mu = 0.005$ and limits of agreement LoA $= \mu \pm 0.012$, see Fig. 3b, and high correlation (Pearson's $r = 0.997$, $p = 0$). Reliability of SC node degree and FC node degree was also very high, with a test–retest and out-of-sample correlation of $r = 0.995$, $p = 0$ and $r = 0.999$, $p = 0$ for FC node degree and $r = 0.998$, $p = 0$ and $r = 0.999$, $p = 0$ for SC degree, respectively, see Supplementary Fig. 4.

**Age, sex and cognition have region-specific, significant associations with SC–FC coupling.** We used a generalized linear model (GLM) to quantify the association between different characteristics of interest and SC–FC coupling. Specifically, subjects' age, sex, total composite cognition score, years of education, intracranial volume (ICV), in-scanner head motion as well as the

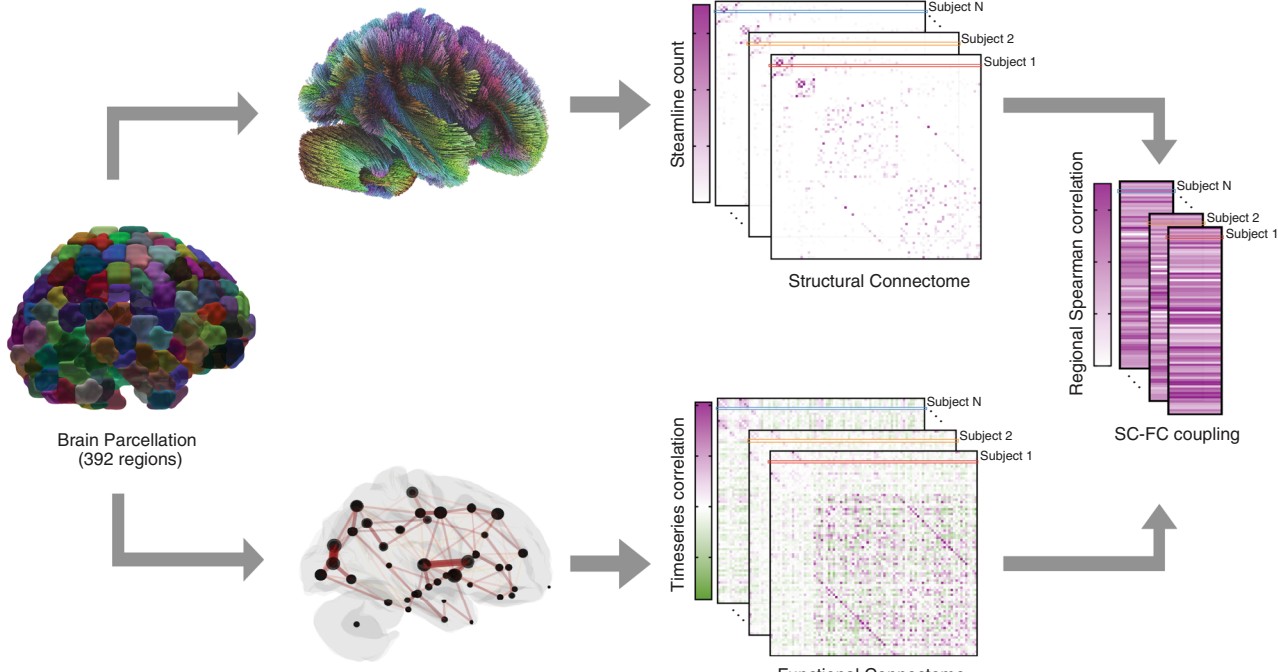

**Fig. 1 Workflow for quantifying regional SC–FC coupling.** The CC400 atlas was used to parcellate the gray matter into 392 cortical and subcortical brain regions[33]. SC matrices were constructed based on probabilistic tractography aimed at reconstructing white-matter pathways. FC matrices, representing similarity of functional activation over time, were computed by correlating average BOLD time series from the defined region pairs. For each subject, corresponding rows in the SC and FC matrices were correlated via Spearman-rank to obtain that region's SC–FC coupling value. The result is a vector of regional SC–FC coupling, of length 392, for each individual.

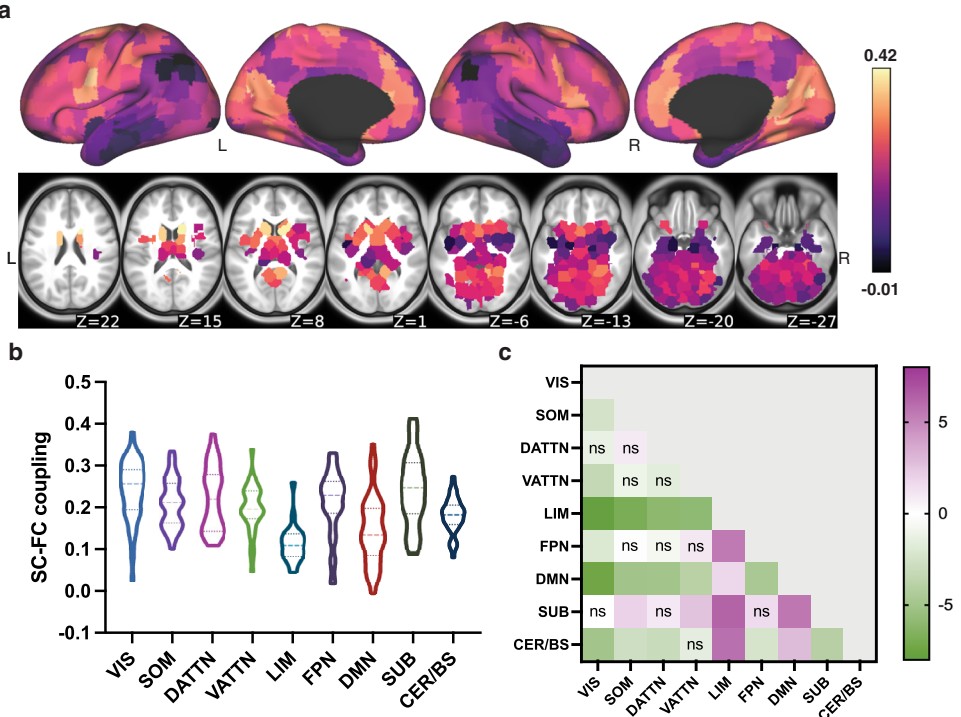

**Fig. 2 Regional whole-brain SC–FC coupling varies spatially across the brain. a** displays the SC–FC coupling for each cortical and subcortical region in the CC400 atlas. **b** shows the distribution of SC–FC coupling over regions grouped into nine different networks (the seven cortical networks defined by Yeo et al.[69], subcortical and cerebellum/brainstem). **c** shows the t-statistics for all pairwise comparisons of SC–FC coupling across networks, calculated as the network on the y-axis versus the network on the x-axis. One-sided p-values were calculated (see detailed description in the "Methods" section). Those comparisons with FDR corrected $p > 0.05$ are marked with ns. Visual and subcortical networks have higher SC–FC coupling than other networks while limbic and default mode areas have weaker SC–FC coupling than other networks. VIS visual, SOM somatomotor, DATTN dorsal attention, VATTN ventral attention, LIM limbic, FPN frontoparietal, DMN default mode, SUB subcortical, CER/BS cerebellum and brainstem.

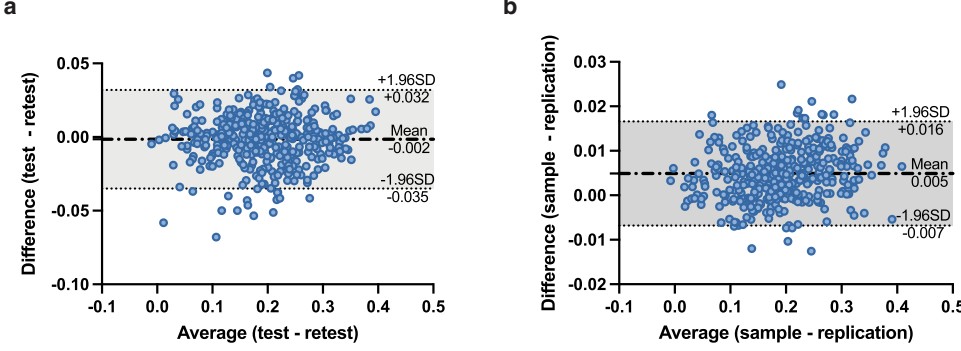

**Fig. 3 Test–retest and sample-replication results show good reliability and reproducibility of SC–FC coupling. a** Bland–Altman plot shows good agreement between the SC–FC coupling calculated in the same set of 41 subjects across two MRI scans taken 6 months apart (mean difference $\mu = -0.002$ and limits of agreement LoA $= \mu \pm 0.034$). **b** Bland–Altman plot shows good agreement between the SC–FC coupling calculated from the original set of 420 subjects and another out-of-sample set of 346 subjects (mean difference $\mu = 0.005$ and limits of agreement LoA $= \mu \pm 0.012$).

two-way interactions terms of age*total cognition score, sex*total cognition score, education*total cognition score and ICV*motion were included in the model. Significant positive associations with age were found in bilateral medial orbito-frontal regions, which belong to default mode network. Significantly negative associations with age were found in the cerebellum (see Fig. 4a, b, and c). Males generally had higher SC–FC coupling than females, with right orbito-frontal gyrus showing largest differences; females had higher SC–FC coupling in right hippocampus (Fig. 4d, e, and f). Higher composite cognition scores were related to decreased SC–FC coupling in bilateral middle cingulate cortex and

supplementary motor area and increased SC–FC coupling in right insula (Fig. 4g, h, and i). There were a mix of positive and negative associations found between SC–FC coupling and in-scanner head motion (see Supplementary Fig. 5); no other covariates in the GLM model had significant relationships with SC–FC coupling.

**SC–FC coupling is heritable and different from FC or SC heritability.** Next, we assessed the heritability of SC–FC coupling using a recently developed modeling approach that considers the level of measurement error of the imaging biomarker in question[26]. Specifically, a linear mixed effect (LME) model was

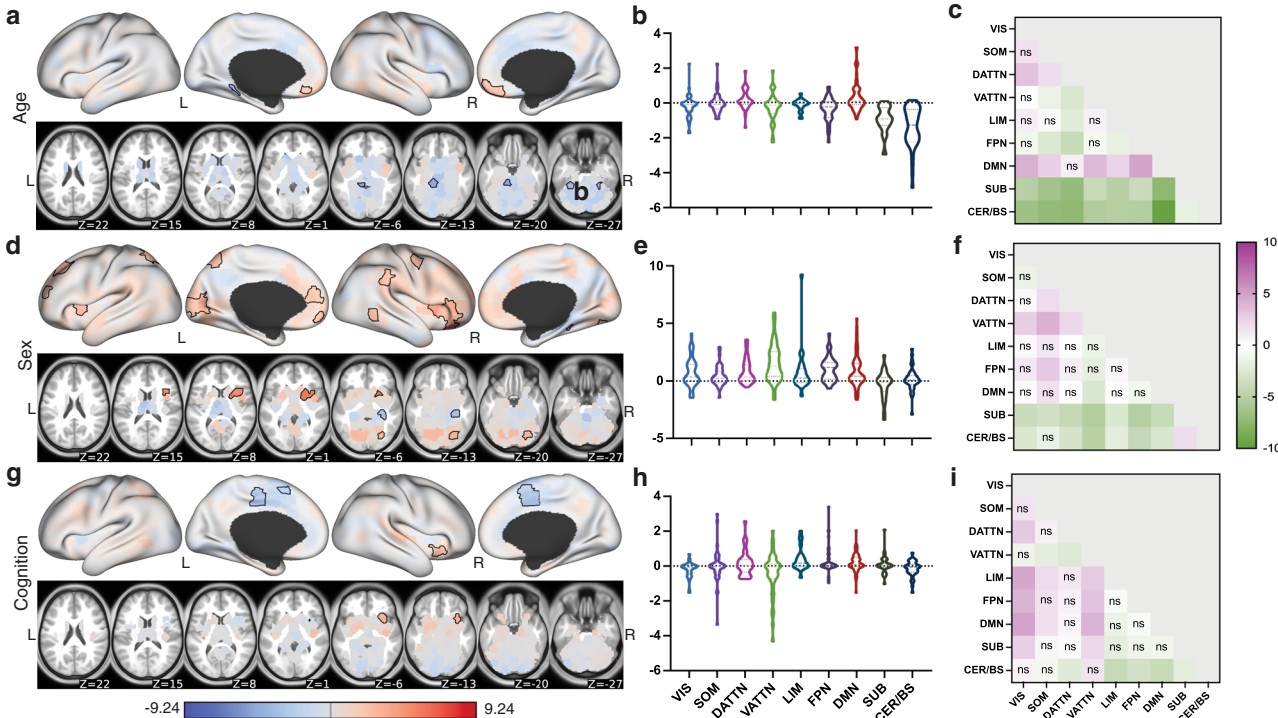

**Fig. 4 Associations between regional SC–FC coupling and age, sex, and total cognition. a**, **d**, and **g** display regional β-values from the GLM quantifying associations between SC–FC coupling and age, sex (blue indicates higher SC–FC coupling in females, red higher in males) and total cognition scores, respectively. Areas with significant β-values (after correction) are outlined in black. **b**, **e**, and **h** show the network-wise β-values for age, sex and total cognition, respectively. **c**, **f**, and **i** show the t-statistics for all pairwise comparisons of the associations across networks, calculated as the network on the y-axis versus the network on the x-axis. One-sided p-values were calculated (see detailed description in the "Methods" section). Those comparisons with FDR corrected p > 0.05 are marked with *ns*.

designed to independently estimate the inter- and intra-subject variation (representing the unstable, transient component and measurement error) of the total phenotype variability. Heritability was defined as the proportion of inter-subject variation attributable to genetics. In addition to age, sex and handedness, we included SC and FC node strength ($l1$ norm of each row) as fixed-effect covariates in the model. Overall, SC–FC coupling was highly heritable, particularly in subcortical, cerebellum/brainstem areas and visual network where the heritability is significantly higher than other networks (median heritability 0.78 ± 0.16, 0.70 ± 0.22, and 0.57 ± 0.20, respectively), see Fig. 5a, b). SC–FC coupling strength was weakly correlated with its heritability (Pearson's $r = 0.114$, $p = 0.140$, see Fig. 5j), suggesting that SC–FC coupling heritability is not associated with its magnitude. For comparison to coupling heritability, we calculated the heritability of each modality's regional node strength, see Fig. 5d, g, with age, sex and handedness as covariates. FC had similar levels of heritability compared to SC–FC coupling, while SC had lower overall levels of heritability. SC–FC coupling heritability was not strongly associated with either SC or FC heritability, as evidenced by the moderate positive correlation between SC–FC coupling and FC heritability (Pearson's $r = 0.309$, $p = 0$) and lack of correlation between SC–FC coupling and SC heritability (Pearson's $r = 0.021$, $p = 0.392$), see (Fig. 5k, l). Further, FC node strength heritability was not correlated with SC node strength heritability (Pearson's $r = 0.086$, $p = 0.089$). The variance explained by each component (genetic effect, common environmental effect, unique environmental effect and intra-subject measurement error) in the heritability models of SC–FC coupling, FC and SC node strength are shown in Supplementary Figs. 13, 14, and 15.

**Sensitivity analyses**. We performed several sensitivity analyses to verify the robustness of the SC–FC coupling results to choices in data processing, atlas definition and method of calculating SC–FC coupling. First, we recalculated SC–FC coupling using anatomically derived 191 region atlas from FreeSurfer[34] (Supplementary Fig. 6); the coupling values appear very similar to the main SC-FC results as do the results of the GLM analyses (Supplementary Fig. 7). We also see good agreement with the main SC–FC coupling values when using FC derived (1) without global signal regression (see Supplementary Fig. 8) and (2) using partial correlation (precision) (Supplementary Fig. 9). Biases in tractography algorithms exist, including the effect of distance between regions, which we adjusted for somewhat using a global filtering approach[35]. SC–FC coupling calculated using partial Spearman-rank with distance between pairs of regions' centroids as a covariate show similarities with the main coupling results (Supplementary Fig. 10). One noticeable difference between the two coupling calculations was weaker subcortical SC–FC coupling when distance was considered in the calculation. We hypothesize this is due to the fact that subcortical structures are further from the majority of cortical regions but also highly connected to all of them so covarying for distance has a greater impact on its coupling measures. It is also known that tractography algorithms underestimate cross-hemisphere connections; SC–FC coupling within a single hemisphere was very similar to whole-brain SC–FC coupling (Supplementary Fig. 11), indicating minimal effects of the under-estimated inter-hemispheric SC on the coupling calculations. Finally, we observe that the varied race/ethnicity of the 941 individuals does not have much influence on heritability estimates; a subgroup analysis of 645 white, non-Hispanic

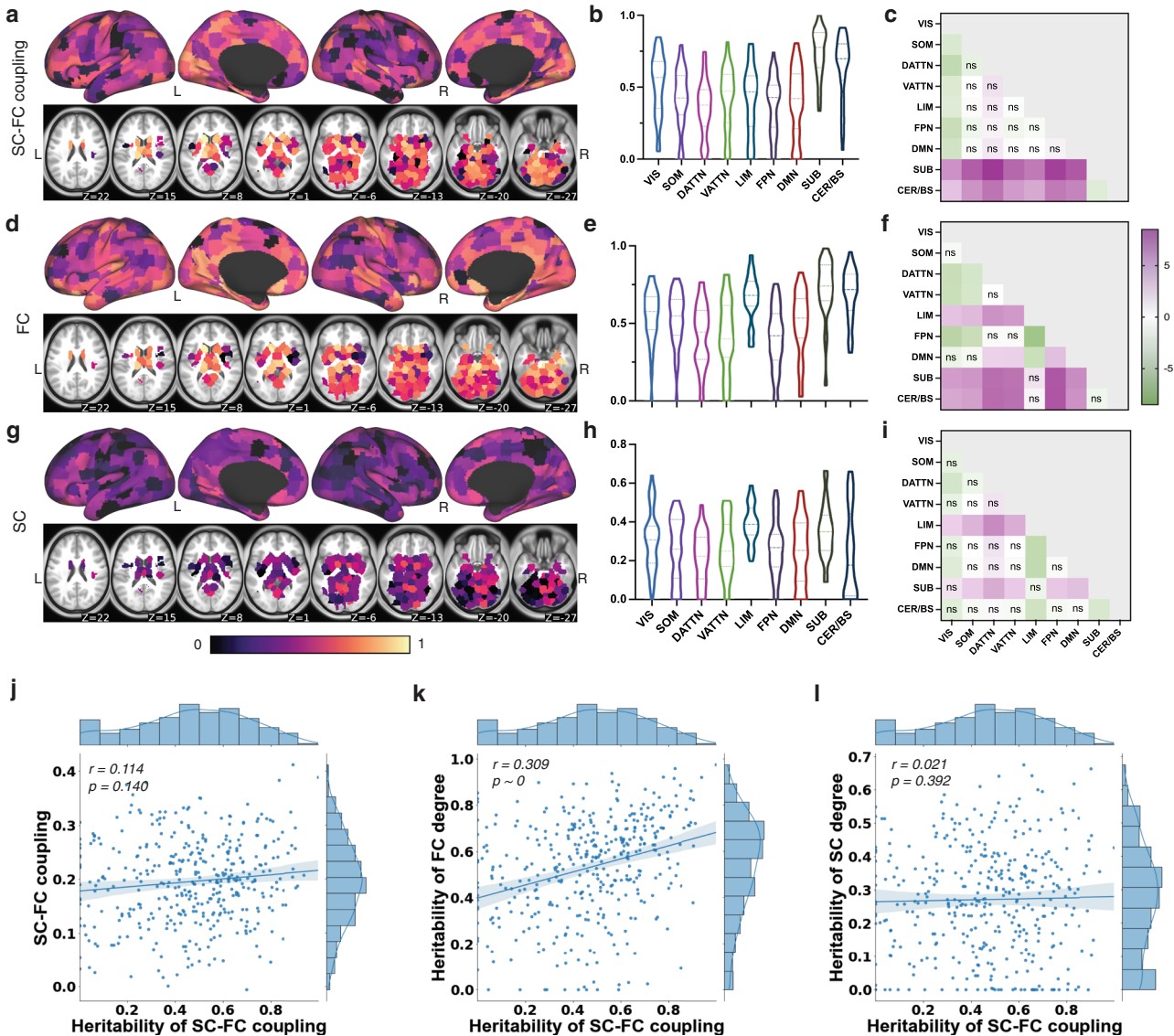

**Fig. 5 SC–FC coupling heritability estimates. a**, **d**, and **g** Regional heritability estimates of SC–FC coupling, SC node strength and FC node strength. **b**, **e**, and **h** Regional heritability estimates of SC–FC coupling, grouped by functional network, for SC–FC coupling, SC node strength and FC node strength, respectively. **c**, **f**, and **i** show the *t*-statistics for all pairwise comparisons of heritability across networks, calculated as the network on the *y*-axis versus the network on the *x*-axis. One-sided *p*-values were calculated (see detailed description in the "Methods" section). Those comparisons with FDR corrected *p* > 0.05 are marked with *ns*. **j** SC–FC coupling heritability has a weak, positive correlation with its signal strength (Pearson's *r* = 0.114, uncorrected *p* = 0.140). **k** and **l** Regional heritability estimates of SC–FC coupling are moderately positively correlated with regional heritability of FC node strength (Pearson's *r* = 0.309, uncorrected *p* = 0) and are not correlated with regional heritability of SC node strength (Pearson's *r* = 0.021, uncorrected *p* = 0.392). The translucent bands around the regression line represent 95% confidence interval for the regression estimate.

individuals revealed consistent heritability patterns in SC–FC coupling (Pearson's *r* = 0.901, *p* = 0), see Supplementary Fig. 12.

## Discussion

In this paper, we quantified the strength of coupling between the structural and functional connectivity profiles of cortical, subcortical and cerebellar brain regions in a large sample of healthy young adults. We demonstrate that SC–FC coupling is strongest in visual and subcortical areas, weakest in limbic and default mode network regions and is consistent across time and different sample populations. Furthermore, we show SC–FC coupling has a positive association with age in bilateral orbito-frontal regions and a negative association with age in the cerebellum, is generally stronger in males, and that stronger SC–FC coupling in the right

insula and weaker coupling in bilateral middle cingulate and supplementary motor areas are related to higher total composite cognition scores. Finally, we show SC–FC coupling is highly heritable, particularly in regions within visual cortex, subcortex and cerebellum/brainstem.

The ordering of cortical regions into anatomical hierarchies, wherein primary sensory regions are at the bottom and higher-order association areas are at the top, provides a way to organize brain regions. Anatomical hierarchies defined by myelination and white-matter connectivity patterns have been shown to reflect functional and transcriptome specialization[36–38]. The cortical SC–FC coupling pattern found in our young adult population, which tracks somewhat with SC degree (see Supplementary Fig. 3), further supports the argument that regional SC–FC coupling may be reflective of anatomical hierarchies[24]. Lower-

order regions of the visual network that have high-cortical myelination and stronger SC node degree tended to have functional activation patterns strongly aligned to their white-matter connectivity profiles. Subcortical structures with the highest SC node degree and lowest FC node degree (see Supplemental Fig. 3) also had very high SC–FC coupling, possibly indicating these regions' roles as relay stations for functional signals traveling between cerebellar, sensory and other cortical regions. Higher-order association areas with lower myelination and weaker SC node degree tend to have complex, dynamic functional profiles that are less anchored by their structural connectivity profiles. Limbic structures that have lower signal-to-noise ratio due to MR imaging artifacts[39] may as a result have weaker SC and FC node degree and SC–FC coupling. Finally, whole-brain SC–FC coupling appeared to be more associated with between-network coupling than within-network coupling. This is likely because of the larger overlap in regions included in the between-network calculation. One issue with calculating the within- and between-network coupling is that the number of regions in the CC400 atlas assigned to each of the 7 Yeo networks is not equal (range: 22−79). Thus, the within and between-network coupling is biased and likely noisy for networks that have a smaller number of regions than ones with a larger number of regions, which complicates comparison.

Functional activation flows not only through direct SC but also indirect, multi-synaptic white-matter connections, which likely contributes to divergence of SC and FC to varying degrees[40]. Statistical, communication, biophysical and machine learning models have been applied to better align FC and SC[3,7,8,41]. Recent work has also found the strength of global SC-FC correlation depends on how FC is calculated[42]. In particular, that work showed FC calculated using partial correlation (precision), which aims to isolate direct and remove the effect of indirect functional connections, had stronger correlations with SC than standard FC calculated using full (Pearson) correlation. However, this observation was based on using Pearson correlation to assess global similarity of the upper triangular portions of the SC and FC matrices, which may not be an appropriate measure as SC is non-Gaussian. In fact, our analyses confirmed that using precision-based FC resulted in higher SC–FC coupling than correlation-based FC, but only when using Pearson correlation to measure SC–FC coupling. When using the more statistically appropriate Spearman correlation to assess the similarity of SC and FC, precision-based FC gives lower values (about half the magnitude) compared to correlation-based FC (see Supplementary Fig. 9). We hypothesize this reduction in coupling may be driven by non-overlapping sparsity patterns that exist in both the SC and the precision-based FC.

Despite the limited age range of our sample (22–37 years) we still observed a few associations between SC–FC coupling and age, with stronger medial orbito-frontal SC–FC coupling and weaker cerebellar coupling being related to increased age. Processes like synaptic pruning, functional diversification and myelination that may impact SC–FC coupling, and are classically associated with adolescent populations, are still occurring in young adults through at least the mid-20s. Orbito-frontal regions of the prefrontal cortex, particularly important in impulse control, are among the last regions in the brain to fully develop[43,44]. Interestingly, Baum et al.[24] found mostly age-related increases (including in medial orbito-frontal regions in agreement with our current findings) and some decreases in SC–FC coupling with increased age during adolescence. Their age-related associations were indeed much more widespread than our findings in young

adults, indicating, unsurprisingly, more dynamic SC–FC coupling in adolescence that continues in some prefrontal regions into young adulthood. We also show sex differences in SC–FC coupling, with males generally having stronger coupling, particularly in right orbito-frontal, default mode and ventral attention networks. Females have higher coupling only in right hippocampus/parahippocampal gyrus. This disagrees with recent findings in young adults that females had overall greater SC–FC coupling than their male counterparts, particularly in left inferior frontal gyrus, left inferior parietal lobe, right superior frontal gyrus and right superior parietal gyrus[25]. They furthermore found higher SC–FC coupling in males in right insula, left hippocampus and right parahippocampal gyrus[25]. Both studies did agree on males having larger SC–FC coupling in right supramarginal gyrus and right insula, but the rest of the results diverge. We hypothesize this may be due to differences in sample size/characteristics or imaging acquisition/preprocessing strategies; particularly important when investigating sex differences is consideration of brain volume and subject motion. Unlike[25], our GLM framework controlled for covariates like in-scanner motion and intracranial volume, which have known sex differences and a complex relationship with BOLD signals[45,46].

Most previous publications investigating SC-FC relationships and their cognitive implications have explored correlations between impairment or cognition with the strength of the correlation between global, whole-brain SC and FC[19,22,47,48]. Studies in control populations have revealed worse cognitive performance in healthy aging was associated with longer latency in dynamic FC states that are more similar to SC[48] and that cognitive flexibility was associated with FC's alignment with SC[22]. Studies in individuals with neurological disorders have shown that SC-FC similarity increases with dementia diagnosis and individuals' performance on memory tasks[47] and that increasing awareness levels in individuals with disorders of consciousness are related to longer latency in dynamic FC states less similar to SC[19]. Regional SC–FC coupling was found to be differently correlated with cognitive function in females and males; specifically, poorer working memory in females was related to weaker SC–FC coupling in local (non-hub/feeder) connections and better reasoning ability in males was related to stronger SC–FC coupling in rich-club hub connections[25]. In their adolescent population, Baum et al.[24] found mostly positive correlations between executive function and SC–FC coupling, particularly in rostro-lateral frontal and medial occipital regions; the only region to show the negative associations with cognitive scores was the right primary motor cortex. In agreement with their findings, we also observe a negative association of regional SC–FC coupling in supplementary motor areas (as well as middle cingulate) with total cognition scores. We also observe positive correlations between SC–FC coupling in right anterior insula/putamen, a region very nearby the rostro-lateral prefrontal area identified in Baum et al., indicating stronger coupling in this area was related to better total cognition scores. The insula is a center of integration of many different domains of brain function; a meta-analysis of the function of the insula revealed an anterio-ventral social-emotional region, a mid-posterior sensorimotor region, a central olfacto-gustatory region, and an anterior-dorsal cognitive region[49]. The anterior insula region we found to have associations between SC–FC coupling and total cognition score overlaps most with the cognitive and social-emotional regulation portions of the insula. Stronger agreement in structure and functional connections in such a highly functionally diverse part of the brain that balances internal states with external environmental responses

could indicate a better coordination of unimodal and transmodal systems.

In this work, we show that regional SC–FC coupling is highly heritable across the brain, particularly in subcortical, cerebellar/brainstem and visual networks. Measurement noise in subcortical regions is highest among the networks, which may suggest increased uncertainty in those regions' heritability estimates (see Supplemental Fig. 13). We find that regional SC–FC coupling heritability is of similar magnitude to FC heritability, and that both are more heritable than SC. Furthermore, we saw that SC–FC coupling heritability was not substantially explained by either SC or FC node strength heritability; in fact, it was only moderately correlated with FC node strength heritability and not correlated with SC node strength heritability. Previous studies have shown heritability of FC profiles, with the default mode network having highest heritability (estimates ranging from 0.42 to 0.8)[26,50]. Our results showed heritability of FC degree in default mode network was indeed significantly higher than other higher-order cortical networks, but not significantly different from visual or somatomotor networks and significantly lower than limbic, subcortical and cerebellar/brainstem networks. Some discrepancy with earlier work may arise from the fact we were measuring heritability of node degree rather than pairwise connections as well as differences in the model used to estimate heritability. Limbic regions in particular had highest heritability among the cortical networks for FC node strength, which contradicts some previous work. However, we observe that the total amount of variance explained by genetics and common/individual environment were lowest and the standard error of the fraction of total variance explained by genetics and common/individual environment were the highest in the limbic network (see Supplemental Fig. 14), indicating possible increased uncertainty in those regions' heritability estimates. From the reliability analysis, it does not appear that the SC's lower heritability values are due to increased measurement noise, as SC node strength was as reliable as FC and SC–FC coupling. Note, however, that since we only have one SC measurement per subject, our approach can not account for with-subject measurement error when estimating the heritability of SC, which might explain some of the differences compared to FC and SC–FC coupling. Previous work has suggested different genetic signatures underlying brain anatomy and physiology[50]; here, heritability of the two modalities' node strengths were indeed not correlated. One recent study quantifying anatomical heritability of the size of cortical areas (as defined by FC) showed unimodal motor/sensory networks had higher heritability (0.44) relative to heteromodal association networks (0.33)[51]. We do observe partial agreement with their findings in that unimodal visual networks, but not somatomotor networks, had higher anatomical SC heritability compared to many other cortical networks.

The results of the analyses in this work are limited by the characteristics of the individuals in the HCP young adult data set. As seen in previous work, SC–FC coupling relationships may vary differently with age across the lifespan, so interpretations of our current findings should be restricted to young adult populations. There are also limitations in the imaging modalities themselves that should be discussed. Motion is an important confound in fMRI and must be mitigated as much as possible; in addition to motion correction and global signal regression, we performed censoring of high-motion frames, which has been shown to further mitigate these effects[52] and included motion as a covariate in the GLM analysis. Tractography algorithms are known to produce streamlines that are not fully reflective of actual anatomical connections[53,54]. Here, we somewhat mitigate this effect by using a global filtering algorithm, which has been shown to result in streamlines that are more reflective of underlying anatomy[35].

Measuring cognition is not an easy task; we chose here to investigate the highest-level composite score (total cognition) but future work could explore more specific cognitive scores like crystallized and fluid intelligence. Furthermore, in this whole-brain, atlased-based analysis of SC–FC coupling, all connections and regions are treated identically, even those in the cerebellum, subcortex and brainstem. We believe that these regions play a very important role in overall patterns of brain activity and white-matter connections so we included them here; however, we also acknowledge that their microanatomy and anatomical connection type (inhibitory versus excitatory) may differ from that of cortical regions. Future work may attempt to modify the SC–FC coupling measure to account for these differences, e.g., treating inhibitory connections differently from excitatory connections. Finally, the approach we used to estimate heritability assumes levels of genetic similarity based on kinship, as classically implemented[26], instead of the more recent approaches that use geneotype data. These recent methods rely on genetic similarity estimates derived from genotype data and thus can be more refined than estimates based on average family relationships. However, genotype-based heritability today is typically computed based on common SNPs and do not account for rare alleles and other types of genetic variation not correlated with common SNPs. Future work will incorporate geneotype data to extend the current estimates of SC–FC coupling heritability.

Understanding how macroscopic anatomical and physiological connectomes are intertwined and can influence behavior or be influenced by an individual's characteristics or environment is an important, unanswered question in human neuroscience. Here, we use neuroimaging, demographic/familial relationship information and cognitive measures in a large population of young healthy adults to begin to uncover some of these associations. We show that regional structure–function coupling is strongest in highly structurally connected visual and subcortical regions, varies with age and sex, is related to composite cognitive scores and is highly heritable. Taken together, these results demonstrate that investigating structure–function relationships at a macroscopic scale can reveal important knowledge in the study of brain form and function.

## Methods

**Data description**. The data for this study comes from the publicly available HCP database containing high-resolution, preprocessed anatomical, diffusion and resting-state functional MRI data. The informed consent for all subjects was obtained by HCP. Our data usage was approved by HCP, and complies with all relevant ethical regulations for work with human participants. Specifically, we use WU-Minn HCP minimally processed S1200 release, which includes high-resolution 3T MR scans, demographics, behavioral and cognitive scores for a population of 1113 young healthy adults (age 22 to 37 years). For the SC–FC coupling results shown in Fig. 2, we used the subset of 420 unrelated subjects that had all four fMRI scans and a complete dMRI scan. Forty-one subjects in HCP had a second MRI scan ~6 months after the first scan (test–retest). The replication (out-of-sample) analysis used another subset of 346 unrelated HCP subjects (age, 28.78 ± 3.80 years; 148 males and 198 females), distinct from the initial set of 420 unrelated subjects. It should be noted that, while each set of subjects did not contain relatives within them, there may be some familial relationships across the two sets of subjects, which could result in an overestimation of the out-of-sample reliability. For the GLM analyses shown in Fig. 4, we took the 415 subjects from the unrelated set of 420 that had total composite cognitive scores (age, 28.69 ± 3.69 years; 213 males, 202 females). For the heritability analysis shown in Fig. 5, we analyzed 941 subjects (age, 28.67 ± 3.70 years; 441 males, 500 females) from 425 different families. In this set of 941 subjects that had all four fMRI scans and a dMRI scan, there were 116 MZ twin pairs, 61 DZ twin pairs, 455 full siblings and 132 singletons (single-birth individuals without siblings).

**Construction of the structural connectomes**. HCP subjects were scanned on a customized Siemens 3T "Connectome Skyra" housed at Washington University in St. Louis. The HCP diffusion data (1.25 mm isotropic voxels, TR/TE = 5520/89.5ms, 3x multiband acceleration, $b$ = 1000, 2000, 3000, 90 directions/shell, collected with both left-right and right-left phase encoding) were first minimally preprocessed by the HCP consortium to correct for motion, EPI and eddy-current

distortion, and registered to each subject's T1 anatomical scan[55]. A multi-shell, multi-tissue constrained spherical deconvolution (CSD) model was computed in MRtrix3 to estimate the orientation distribution function[56]. We used a probabilistic (iFOD2[57]), anatomically constrained (ACT[58]) tractography algorithm with dynamic white-matter seeding to create individual, whole-brain tractograms containing 5 million streamlines for each subject. To better match the whole-brain tractogram to diffusion properties of the observed data, we also computed streamline weights that are designed to reduce known biases in tractography data (SIFT2[35]). Finally, the tractograms were used to estimate SC weights for the CC400[33] atlas. The SC between any two regions was the SIFT2-weighted sum of streamlines connecting those regions divided by the sum of the gray matter volume of those regions. The result was an ROI-volume normalized pairwise SC matrix for each subject.

**Construction of the functional connectomes**. There were four gradient-echo EPI resting-state fMRI runs (2.0 mm isotropic voxels, TR/TE = 720/33.1 ms, 8x multiband acceleration, FoV = 208 × 180 mm$^2$, FA = 52°, 72 slices) of ~15 min each, with two runs in one session and two in a second session, where each session included both right-left and left-right phase encoding. There were 1200 volumes for each run and a total of 4800 volumes (1200 volumes × 4 runs) for each subject. The data were minimally preprocessed[55] and ICA + FIX[59,60] denoised by the HCP consortium[61]. For each time series, motion and global signal outlier timepoints were identified using an approach adapted from the Artifact Detection Tools (ART) from the CONN Toolbox[62]. Motion outliers were identified by applying motion parameter estimates to a set of 6 control points at the face centers of a 140 × 180 × 115 mm brain-sized bounding box, and selecting all timepoints where any face center moved by more 0.9 mm. Global signal outliers were identified by computing the temporal derivative of the global mean time series across the brain, prior to any additional temporal filtering aside from a linear detrending, and selecting timepoints where this temporal derivative deviated from the temporal mean by 5 standard deviations. Timepoints that met any of these outlier conditions, as well as their neighboring timepoints, as well as the first 10 volumes from each scan, were ignored during subsequent processing and analysis. Additional nuisance regressors included an offset term, linear trend, 6 motion parameters and their derivatives, squares, and squared derivatives (24 motion regressors), and 10 Anatomical CompCor (aCompCor) regressors to reduce the contribution of signals related to white matter and CSF (5 principal components from each, using FreeSurfer-derived masks eroded by 2mm). Simultaneous with the nuisance time series regression, we regressed out the effect of global gray matter signal and its temporal derivative[63]. Outlier-free temporal filtering was performed after nuisance regression, using a discrete cosine transform (DCT) projection filter. Outlier-free correlation analyses ignored the censored timepoints. In scanner motion for each individual was quantified by averaging the overall frame-wise displacement for each of the four fMRI scans. FC matrices Σ were calculated using the Pearson correlation between each region-pair's average time series in the CC400 atlas[33], resulting in four FC matrices for each subject. For all the analyses except heritability, the four FC matrices were averaged together. The heritability analysis uses each of the individual's four scans independently to incorporate between-measurement variability into its estimates of heritability[26].

**Calculation of SC–FC coupling**. SC–FC coupling was constructed by calculating the Spearman-rank correlation between a row of the SC matrix with the corresponding row of the FC matrix (excluding the self-connection). The result of this step in the analysis is, for each individual, a vector of length 392 that represents the regional SC–FC coupling strength, or structure–function alignment, for each of the 392 regions in the atlas. We chose non-parametric Spearman-rank correlation to quantify the similarity of a region's structural and functional connectivity pattern to the rest of the brain as it is a measure that is straightforward and easily interpreted and, importantly, accommodates the non-Gaussianity of the entries in the SC. In addition, we wanted to compare the results found here in young adults to previous work using a similar approach in adolescents wherein Spearman-rank correlation was used to quantify SC-FC alignment[24]. To assess the association of between and within-network coupling to whole-brain coupling, we separately calculated, for each region, its between and within-network SC–FC coupling as follows. Within-network SC–FC coupling for each region was the Spearman correlation of the structural and functional connections between that region and other regions in the same network; between-network SC–FC coupling was the same calculation but between that region and regions outside of it's assigned network. To compare these two network-specific measures to whole-brain SC–FC coupling, we calculated Pearson correlation between the measures; p-values were calculated using a permutation test with 10,000 resamples.

We also performed several ancillary analyses to verify the robustness of our SC–FC coupling results to choices in data processing, atlas definition and method of calculating SC–FC coupling. To validate the main findings with the functionally defined CC400 atlas, we also used an anatomically derived 191 region atlas from FreeSurfer, with 148 cortical regions from Destrieux + 16 subcortical regions from FreeSurfer's aseg volume and 27 cerebellar regions from SUIT. We also included two additional versions of FC: one without global signal regression and one calculated using partial correlation, or precision. It is known that there are biases that exist in tractography algorithms, specifically in the effect of distance between

regions. Therefore, we also calculated SC–FC coupling using partial Spearman-rank with distance between region-pair centroids as a covariate. Finally, it is known that tractography algorithms underestimate cross-hemisphere connections; therefore we also calculated SC–FC coupling within a single hemisphere for comparison to the whole-brain SC–FC coupling measure.

**Interpretation of statistical measures**. We constructed violin plots in each figure to demonstrate the distribution of the various measures across nine different networks. The median of each distribution is represented with a dashed line and the quartiles are represented using dotted lines; the shape of the violin is representative of the underlying data. Pairwise comparisons were done within the networks and the heatmaps in each figure show the unpaired t-statistic comparing the network values. Significance of the t-statistic and correlations were quantified using BrainSMASH (Brain Surrogate Maps with Autocorrelated Spatial Heterogeneity)[64], which was developed for statistical testing of spatially auto-correlated brain measures. All p-values (for t-tests or correlations) were computed using BrainSMASH to generate 1000 surrogate maps and then counting the ratio of these surrogate maps having values more extreme than the original t-statistic or correlation (one-sided p-values).

Reliability of SC–FC coupling, SC node strength and FC node strength was assessed by calculating Pearson correlation between the three measures extracted from the test and retest visits (N = 41) and between the measures extracted from the original sample (N = 420) the out-of-sample population (N = 346). Bland–Altman plots were also used to quantify the reliability of SC node strength, FC node strength and SC–FC coupling, which gave us LoA for each of the measures. The mean difference, also called the bias, is calculated by

$$\bar{d} = \frac{1}{n}\sum_{i=1}^{n} d_i \tag{1}$$

and the LoA between the test–retest and out-of-sample replication studies are defined by a 95% prediction interval of a particular value of the difference, which are computed as

$$\bar{d} \pm 1.96 S_d \tag{2}$$

where $S_d = \sqrt{\frac{1}{n-1}\sum_{i=1}^{n}(d_i - \bar{d})^2}$.

**Quantifying relationships between SC–FC coupling, age, sex, and cognition**. There are several different covariates that we hypothesized may have significant relationships with SC–FC coupling, namely, age, sex, years of education, total cognition score, intracranial volume (ICV) and in-scanner head motion. The Total Cognition score, measured using the tests in the NIH toolbox, is the average of the crystallized score (including Picture Vocabulary and Reading Recognition measures) and fluid score (including Dimensional Change Card Sort, Flanker Inhibitory Control and Attention, Picture Sequence Memory, List Sorting, and Pattern Comparison measures). To calculate in-scanner head motion for each subject, we averaged the frame-wise displacement over each volume in the fMRI time series, and then took the average across the four fMRI scans. Finally, using a generalized linear model (GLM) approach, we assessed regional associations between SC–FC coupling and in-scanner motion, demographics and cognitive scores, plus four interaction terms (age*cognitive score, sex*cognitive score, years education*cognitive score, and ICV*motion). The four interaction terms we included in the GLM were those pairs of variables that we hypothesized may have non-negligible interactions.

$$y_k = \beta_0 + \sum_{i=1}^{10} \beta_i x_i \tag{3}$$

where $y_k$ is the SC–FC coupling of length $n$ (number of subjects) for region $k = 1$, 2, . . . 392, $\beta_0$ is the intercept and $\beta_i$ are the coefficients for each covariate $x_i$, also a vector of length $n$. SC–FC coupling values were Fisher r-to-z transformed for improving normality. All p-values for the regression coefficients were FDR corrected for multiple corrections and analyzed for significance at a level of $\alpha = 0.05$.

**Quantifying the heritability of SC–FC coupling**. LME models were developed to disentangle inter- versus intra-subject variation[65]. This LME approach was recently adapted for and applied to HCP data to quantify heritability of functional connectome fingerprints with respect to the inter-subject component, while removing the effect of transient changes across observations of a single subject[26]. This approach allows examination of the association between the genetic relationship and phenotypic similarity, while accounting for shared environment of siblings. Specifically, we write the following:

$$y_{ij} = x_{ij}\beta + \gamma_i + \epsilon_{ij} \tag{4}$$

where $i = 1, 2, . . ., n$ and $j = 1, 2, . . . m_i$. $m_i$ is the total number of repeated measures for subject $i$. The variable $y_{ij}$ is the phenotype measurement for subject $i$ for measurement $j$, $x_{ij}$ contains all the $q$ covariates while the vector $\beta$, also of length $q$, contains the unknown fixed population-level effects. The scalar $\gamma_i$ donates the subject-specific deviation from the population mean and $\epsilon_{ij}$ describes denotes the intra-subject measurement error (transient component) of $y_{ij}$ and is assumed to be independent of the random effects and independent between repeated

measurements. Stacking all subjects and all repeated observations into a single vector, we have

$$\mathbf{y} = \mathbf{x}^T \boldsymbol{\beta} + \mathbf{T}\gamma + \boldsymbol{\epsilon}, \tag{5}$$

where $\mathbf{y}$ is the phenotype vector of length $n_{total} = \sum_{i=1}^{n} m_i$, $\mathbf{x}$ is the covariate matrix of dimension $q \times n_{total}$, $T$ is a block diagonal matrix of dimension $n_{total} \times n_{subj}$, $\gamma$ is a vector of length $n_{subj}$ and $\boldsymbol{\epsilon}$ is a vector of length $n_{total}$. We consider $\gamma$ to be the sum of three different effects: additive genetic effect $\mathbf{g} \sim N(0, \sigma_A^2 \mathbf{K})$, shared (common) environmental effect $\mathbf{c} \sim N(0, \sigma_C^2 \Lambda)$ and unique (subject-specific) environmental effect $\mathbf{e} \sim N(0, \sigma_E^2 \mathbf{I}_{\mathbf{n}_{total}})$. Here, $\sigma_A^2$, $\sigma_C^2$, and $\sigma_E^2$ are the additive genetic variance, common environmental variance and unique environmental variance, respectively. The matrix $\mathbf{K}$ is the $m \times m$ genetic similarity matrix derived from the pedigree information where $K_{ij}$ is 1 for monozygotic twins, 1/2 for dizygotic twins and full siblings and 0 for unrelated individuals. The matrix $\Lambda$ is an $n_{subj} \times n_{subj}$ matrix indicating shared environment, that is, if the two subjects $i$ and $j$ have the same parents then $\Lambda_{ij}$ is set to 1, otherwise it is set to 0. Finally, the matrix $\mathbf{I}_{\mathbf{n}_{total}}$ is the identity matrix of size $n_{subj} \times n_{subj}$. Intra-subject variation is assumed to follow a Gaussian distribution, $\boldsymbol{\epsilon} \sim N(0, \sigma_M^2 \mathbf{I}_{\mathbf{n}_{total}})$. Thus, the covariance matrix of $\mathbf{y}$ is

$$\mathbf{cov}[\mathbf{y}] = \sigma_A^2 \mathbf{TKT}^T + \sigma_C^2 \mathbf{T}\Lambda \mathbf{T}^T + \sigma_E^2 \mathbf{TT}^T + \sigma_M^2 \mathbf{I}_{\mathbf{n}_{total}}. \tag{6}$$

Finally, we can define the non-transient heritability of a given trait as the proportion of stable, non-transient inter-subject variation that can be explained by genetic variation in the population as

$$h^2 = \frac{\sigma_A^2}{\sigma_A^2 + \sigma_C^2 + \sigma_E^2} \tag{7}$$

Unbiased estimates of the variance components $\sigma_A^2$, $\sigma_C^2$, $\sigma_E^2$ and $\sigma_M^2$ were obtained using the restricted maximum likelihood (ReML) algorithm[66]. We estimated the non-transient heritability of regional SC–FC coupling (four measurements per subject), SC node strength as calculated via the sum of rows, excluding the diagonal (one measurement per subject) and FC node strength as calculated via the sum of absolute value of rows, excluding the diagonal (four measurements per subject). SC–FC coupling, FC node degree and SC node degree were standardized before calculating heritability. Age, sex, and handedness were taken as fixed-effect covariates in each of the heritability models; SC node strength and FC node strength were also considered fixed-effects covariates in the SC–FC coupling heritability model. Finally, because there may be differences in genetic similarity patterns across race/ethnicity, we recalculated heritability of the various measures using a homogeneous subset of white, non-Hispanic individuals ($N = 645$).

**Citation gender diversity statement**. Recent work in several fields of science has identified a bias in citation practices such that papers from women and other minorities are under-cited relative to the number of such papers in the field[67]. Here, we sought to proactively consider choosing references that reflect the diversity of the field in thought, form of contribution, gender, and other factors. We obtained predicted gender of the first and last author of each reference by using databases that store the probability of a name being carried by a woman[67]. By this measure (and excluding self-citations to the first and last authors of our current paper), our references contain 7.81% woman(first)/woman(last), 10.94% man/ woman, 20.31% woman/man, 60.94% man/man. This method is limited in that (a) names, pronouns, and social media profiles used to construct the databases may not, in every case, be indicative of gender identity and (b) it cannot account for intersex, non-binary, or transgender people. We look forward to future work that could help us to better understand how to support equitable practices in science.

**Reporting summary**. Further information on research design is available in the Nature Research Reporting Summary linked to this article.

## Data availability
HCP MRI data and most behavioral data are publicly available at https://db. humanconnectome.org/with the acceptance of HCP Open Access Data Use Terms. Some data elements, including family structure, exact age, handedness, ethnicity and race, are available only to qualified investigators who agree to HCP's Restricted Data Use Terms. Applications for access to Restricted Data should be submitted by every investigator who will view and use the data, will be processed individually, and approval is on a case-by-case basis. The time to get the data is about two weeks. Once given the access, researchers should abide by a prohibition against publishing certain types of individual data in combination that could make a person individually recognizable, or that could harm and embarrass someone who was inadvertently identified. The specific data used in this work is WU-Minn HCP Data-1200 Subjects and WU-Minn HCP Retest Data. Source data to plot the figures are provided with this paper. Source data are provided with this paper.

## Code availability
Python code to reproduce the main results of this paper is publicly available at https:// github.com/zijin-gu/scfc-coupling[68]. The original code of the LME model for heritability estimation is publicly available at https://github.com/ThomasYeoLab/ Standalone_Li2019_GSR/tree/master/external_packages/LME. The functional and

diffusion data were already preprocessed by HCP. Example script to run HCP tractography can be found at https://github.com/zijin-gu/scfc-coupling. The denoising, ROI time series extraction, volume censoring, and FC extraction code is publicly available at https://github.com/kjamison/fmriclean.

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

## Acknowledgements

This work was supported by the following grants: RF1 MH123232 (AK), R21 NS104634-01 (AK), R01 NS102646-01A1 (AK), R01 LM012719 (MS), R01 AG053949 (MS), NSF CAREER 1748377 (MS), and NSF NeuroNex Grant 1707312 (MS). Data were provided by the Human Connectome Project, WU-Minn Consortium (Principal Investigators: David Van Essen and Kamil Ugurbil; 1U54MH091657) funded by the 16 NIH Institutes and Centers that support the NIH Blueprint for Neuroscience Research; and by the McDonnell Center for Systems Neuroscience at Washington University.

## Author contributions

A.K. and M.S. conceived the experiments and interpreted the results, Z.G. conducted the experiments, analyzed, and interpreted the results. K.J. processed the imaging data and interpreted the results. Z.G. and A.K. wrote the manuscript. All authors reviewed the manuscript.

## Competing interests

The authors declare no competing interests.
