## [Peer Review File · Nature Communications]

REVIEWER COMMENTS

Reviewer #1 (Remarks to the Author):

This is an interesting paper. Overall, the methods are clearly stated, and the analyses that are described appear well-executed. I have some comments about the theoretical motivation of this work, lack of adequate analyses for motion-related issues that could inflate relationships and mitigate the interpretability of neural effects, and a few other unaddressed methodological choices and statistical issues that could influence the findings. If thoroughly addressed, this work could be suitable for publication.

Specific comments:

Heritability is typically best estimated using genotype-phenotype data. The procedure used here obtains heritability estimates using the assumptions of a previously conducted study. This should be acknowledged in the limitations.

The preprocessing routine is stated to be minimal, and does not appear to apply techniques that more aggressively regress motion parameters and their derivatives to adequately mitigate their influences on intrinsic FC data. The goal of the study appears to be about the brain and not about heritable features of behavioral motion timeseries that can express themselves in neuroimaging data. Thus, it is important to re-analyze the data after applying a more thorough preprocessing routine (e.g., the procedures validated by Ciric et al., 2017, or similar work from Satterthwaite and colleagues or other research groups who have investigated these issues).

Ciric, Rastko, et al. "Benchmarking of participant-level confound regression strategies for the control of motion artifact in studies of functional connectivity." *Neuroimage* 154 (2017): 174-187.

From the prior paper introducing the heritability estimates, roughly 70% of the FC variance was not due to heritable effects. However, the current paper reports that up to 78% of SC-FC coupling is estimated to be heritable. Some of this effect could be attributed to the region/system-level specificity discussed in the current paper. However, it could be helpful to more thoroughly discuss why SC-FC coupling might have higher heritability than FC in some cases (here again, having addressed any changes in estimates after preprocessing the data with more aggressive motion-correction procedures).

Figure 5K shows a strong association between SC-FC coupling and FC heritability. The distribution of FC across regions likely averages to a higher heritability than that found in the 2017 study. Can this be quantified, and if so, why? Ruling out preprocessing and motion issues here also seems important. Further, if FC heritability drives (or is highly related to) SC-FC coupling, does this mitigate interest or relevance of SC-FC coupling as a unique measure? This question is related to my other theoretical and methodological points.

Like most connectomes, the degree and strength distributions of node-wise structural and functional connectivity are likely not perfectly Gaussian, and z-scoring alone will not remove influences like skewness and kurtosis of the distributions. These effects can strongly influence Pearson's correlation-based measures of SC-FC coupling. This should be addressed directly, for instance by applying a specific test of region-wise skewness and kurtosis of row-wise statistical distributions, then selecting the summary statistic accordingly or applying a transformation prior to subsequent analyses.

While the SC-FC measure used here is relatively simple, (row-wise Pearson's correlations between SC and FC elements), it is not the only way to encode this relationship. It emphasizes the connections between a node and all other regions, which does not appear to have a theoretical motivation in the current study. Other techniques such as alternatives to correlation (overlap indices, those that are more appropriate to non-Gaussian distributions), region-wise measures, or spectral decomposition (among others) could be evaluated. In addition, depending on the measure, associations with cognition and behavior can be strongly influenced (e.g., Medaglia et al., 2018). Why was this SC-FC technique used, other than for convenience? What is its theoretical motivation, and what alternatives could be justified?

Medaglia, J. D., Huang, W., Karuza, E. A., Kelkar, A., Thompson-Schill, S. L., Ribeiro, A., & Bassett, D. S. (2018). Functional alignment with anatomical networks is associated with cognitive flexibility. *Nature human behaviour*, 2(2), 156-164.

Further, the analyses do not encode system-wide constraints when computing the SC-FC measures (for instance, only including system-to-system elements in the analysis). This muddies some of the analyses because region-wise and system-wise effects are not well-disentangled. The system-level findings appear to involve nodes with projections to every system, not just system-to-system connections. Further addressing this issue seems justified.

Informalisms on line 269, 271, section starting at 278: the colloquial term "fingerprint" has been popularized, but the brain is not a finger and structural and functional connectivity (and their association) are not prints. Language describing the exact methods and their interpretation would be clearer. For instance, at line 268, the authors state that "coupling was constructed by calculating the Pearson correlation between a row of the SC matrix, representing the connectivity fingerprint of that region in the brain". Omitting the phrase "representing the connectivity fingerprint of that region in the brain" results in a clear statement. In addition, it reduces the impression that the only way to represent structure-function coupling is using row-wise Pearson's correlations between structural and functional data.

Atlas-based analyses often overlook important consequences on the neural data and the corresponding interpretation. For instance, the microanatomy, cell body density, and anatomical connections involving the basal ganglia and cerebellum are quite different from the cortex. How should findings concerning those regions be couched and what appropriate future directions could be pursued?

Reviewer #2 (Remarks to the Author):

In "Regional structural-functional connectome coupling is heritable and associated with age, sex and cognition in adults" Gu et al explore the distribution, reliability, and heritability of regional structure-function coupling in young adults, and examine how it relates to age, sex, and cognition. This paper has a great deal to recommend it, including an interesting and important topic, appropriate use of a large data resource like the HCP, clear writing, excellent visualizations, code availability, and appropriate methods. Overall, I think it makes a valuable contribution and will be of interest to a broad readership. That being said, I have several comments for the authors to consider; I have signed this review for transparency and am happy to discuss these comments with the authors if they are unclear. -Ted Satterthwaite

1. Sample selection. For readers who are not intimately familiar with the HCP, the methods describing the sample selection criteria for the various analyses are somewhat opaque. Inclusion of a flowchart for inclusion with clear specification of QA criteria for all modalities used would be informative.
2. Regularization of FC matrices. The methods for generating the precision matrices was somewhat hard to follow; slowly stepping through these methods (potentially with a schematic) would make this more accessible. For example, how was overfitting avoided against in the grid search / parameter selection process? Was cross-validation used?
3. Test/retest section. I thought this was a very valuable element of the paper, and as such was surprised to see that there was (as far as I can tell) zero description of these procedures in the methods. While this oversight can be easily remedied, it would be additionally quite informative to expand this section and contrast SC/FC coupling with the component measures, like the analyses included in the heritability analyses – i.e., SC, FC matrices. Furthermore, it would be useful to include supplementary analyses evaluating the relative reliability of precision-FC vs. correlation-FC (for both FC itself and SC/FC coupling). This would also potentially help contextualize the heritability analyses – i.e., was low heritability of SC due simply to noise in the SC measures? Does precision FC allow for greater reliability?

4. Distance dependence. Tractography methods have a known distance dependence. While the appropriate preprocessing used may minimize this, additional analyses where inter-node Euclidean distance is treated as a covariate would bolster confidence that the observed results were not due to this bias.

5. Results in discussion. There are several instances of new results being presented in the discussion section. These would be better placed in the results section, with transparent procedures reported in the methods.

6. Comparing regional profiles. There are multiple figures/analyses in the paper where regional profiles are compared – for example, figure 5 comparing heritability of SC to coupling. R values are provided to estimate effect size, and p values seem to be based on parametric statistics. While the analyses are useful, the p values from significance tests are not valid due to the non-independence of measures and the arbitrary atlas resolution; i.e., greater “statistical power” could be achieved by simply using a higher resolution atlas (i.e., 100 nodes vs. 10,000 nodes). Testing procedures that consider these factors should be used—i.e, the “spin test” (<https://www.sciencedirect.com/science/article/abs/pii/S1053811918304968>) or “brains smash” (<https://brainsmash.readthedocs.io/en/latest/>). Given the robustness of the findings, I do not imagine this will substantially alter the results.

Reviewer #3 (Remarks to the Author):

Review: Gu et al., regional structure-functional connectome coupling is heritable and association with age, sex and cognition in adults.

In the current paper Gu and colleagues studied SC-FC coupling, the association between structural and functional connectivity, and its genetic basis, and association with age, sex, and cognitive abilities in the HCP sample. As reported in previous work (Vázquez-Rodríguez, 2019; Baum, 2020), the researchers observed SC-FC coupling was highest in visual and somatomotor areas, and lower in association regions, sub-cortex and cerebellum. Next, the authors show that SC-FC coupling decreases with age, and is associated with sex and cognition. Last, they observed that the coupling of SC-FC is heritable, especially in visual, dorsal attention, and fronto-parietal networks, more than SC or FC alone.

The current insights in structure-function coupling are largely in line with previous papers on this topic such as Vázquez-Rodríguez, 2019 or Baum, 2020. Although the associations with sex and its heritability are interesting, I am not very sure whether the insights gained in the current study extend previous knowledge significantly. I hope my comments can be of help.

To provide more specific comments:

1. Why did the authors perform global signal regression? Did results remain similar when not running GSR?
2. Did the authors consider the lack of cross hemisphere connections using DTI?
3. Why did the authors use the CC400 atlas? Would an anatomically defined or functionally informed atlas not be more appropriate when assessing structure-function relationships?
4. For the association with age, is the sample's age range really appropriate to make statements about ageing? What would aging mean in a young adult sample? Why not use a second sample with a broader age-range for such an evaluation?
5. It is not clear to me what the network-based coupling is often inverse (Figure 2C) between the association from network $X > Y$ versus $Y > X$.
6. Did the authors control for ICV in their analysis of the relationship between sex and/or cognition and SC-FC coupling?
7. For the heritability analyses, it seems that the FC / SC strength is a new and different measure, and I am unsure how this would relate to SC-FC coupling. As such comparing the heritability of FC to SC-FC coupling is unclear to me. Looking at the patterns, aren't heritability values a bit biased

by the signal strength? E.g. subcortical regions and limbic regions have much lower h^2_r and signal strength?

8. Possibly it would help to add standard deviations, measures of variance to the mean values reported in the text.

9. Why is the association with average myelination discussed in the discussion but not in the results?

10. In the discussion it was mentioned the heritability of structural connectivity was presented for the first time. However, this has been reported before in different contexts (such as heritability of T1w/T2w ratio or DTI metrics)

Reviewer #4 (Remarks to the Author):

In this manuscript, Zijin Gu and colleagues describe their investigation of regional structural-functional connectome (SC-FC) coupling in a sample of healthy young adults aged 22 to 37 from the Human Connectome Project (HCP) cohort. They investigate sex and age differences, the relationship with cognitive ability, and estimate heritability.

The study is novel and fills a gap in knowledge in the sense that it investigates regional SC-FC coupling in a large MRI cohort.

Overall, the manuscript is well written and easy to follow, and it is of interest to the broader field of human neuroscience. In my view, the methods are appropriate and well executed. I have a few suggestions/comments for the authors that could help make the paper stronger:

Major point:

- I feel that the paper falls short in terms of the interpretation and providing biological insights.

The authors could expand their discussion of findings in the context of the broader SC and FC literature.

- Heritability is a population-specific parameter. The HCP cohort is ethnically diverse. The number of white non-Hispanic individuals is $N \sim 720$. I would suggest running a sensitivity analysis using only this homogeneous subset of individuals.

- The authors could try to disentangle the relationship between regional SC-FC coupling and crystallized vs. fluid intelligence separately.

- The authors could include years of education in the GLM (e.g., to explore whether the association between SC-FC coupling is somewhat mediated/moderated by education)

Minor points:

- Line 18 - I would suggest changing the term "pathological populations".

- In lines 45-50, the authors talk about heritability and then genetic co-expression. This is a bit confusing.

- Figure 3 could be supplementary.

- In the legend of Figure 5, there is a typo: "Rregional"

Response to Reviewers

Title: Regional structural-functional connectome coupling is heritable and associated with age, sex and cognitive scores in adults

**Manuscript Reference Number:
NCOMMS-20-48944**

Authors:

Zijin Gu

Keith Jamison

Mert Sabuncu

Amy Kuceyeski

Date: April 14, 2021

Response To Reviewer #1

Overall Comments

This is an interesting paper. Overall, the methods are clearly stated, and the analyses that are described appear well-executed. I have some comments about the theoretical motivation of this work, lack of adequate analyses for motion-related issues that could inflate relationships and mitigate the interpretability of neural effects, and a few other unaddressed methodological choices and statistical issues that could influence the findings. If thoroughly addressed, this work could be suitable for publication.

Response

We appreciate your careful review and detailed feedback. We have stated the theoretical motivation of this work, added more detailed description of the fMRI data preprocessing steps, especially our extensive motion correction which was not well described in the original version, and changed the method for calculating SC-FC coupling. We believe the changes have resulted in a much stronger manuscript that represents an important advancement in the literature of SC-FC relationships.

Reviewer Comment

Heritability is typically best estimated using genotype-phenotype data. The procedure used here obtains heritability estimates using the assumptions of a previously conducted study. This should be acknowledged in the limitations.

Response

Thank you for pointing out this potential drawback. We have added discussion of this limitation to the Discussion section:

"Finally, the approach we used to estimate heritability assumes levels of genetic similarity based on kinship, as classically implemented (Ge et al., 2017), instead of the more recent approaches that use genotype data. These recent methods rely on genetic similarity estimates derived from genotype data and thus can be more refined than estimates based on average family relationships. However, genotype-based heritability today is typically computed based on common SNPs and do not account for rare alleles and other types of genetic variation not correlated with common SNPs. Future work will incorporate genotype data to extend the current estimates of SC-FC coupling heritability."

Reviewer Comment

The preprocessing routine is stated to be minimal, and does not appear to apply techniques that more aggressively regress motion parameters and their derivatives

to adequately mitigate their influences on intrinsic FC data. The goal of the study appears to be about the brain and not about heritable features of behavioral motion timeseries that can express themselves in neuroimaging data. Thus, it is important to re-analyze the data after applying a more thorough preprocessing routine (e.g., the procedures validated by Ciric et al., 2017, or similar work from Satterthwaite and colleagues or other research groups who have investigated these issues).

Ciric, Rastko, et al. "Benchmarking of participant-level confound regression strategies for the control of motion artifact in studies of functional connectivity." *Neuroimage* 154 (2017): 174-187.

Response

Thank you for the comments, we agree that motion can have very profound effects on MRI. After re-examining the Methods section, we realized that our description of the pipeline was incomplete. We now have revised it to include all of the steps we took to process and correct the data for motion effects, see Methods section:

"For each time series, motion and global signal outlier timepoints were identified using an approach adapted from the Artifact Detection Tools (ART) from the CONN Toolbox (Whitfield-Gabrieli and Nieto-Castanon, 2012). Motion outliers were identified by applying motion parameter estimates to a set of 6 control points at the face centers of a $140 \times 180 \times 115$ mm brain-sized bounding box, and selecting all timepoints where any face center moved by more 0.9mm. Global signal outliers were identified by computing the temporal derivative of the global mean time series across the brain, prior to any additional temporal filtering aside from a linear detrending, and selecting time points where this temporal derivative deviated from the temporal mean by 5 standard deviations. Timepoints that met any of these outlier conditions, as well as their neighboring timepoints, as well as the first 10 volumes from each scan, were ignored during subsequent processing and analysis. Outlier-free temporal filtering was performed using a discrete-cosine projection filter. Outlier-free correlation analyses ignored the censored timepoints."

We also added some limitations to the discussion section in this regard to emphasize this very important issue in fMRI studies.

"Motion is an important confound in fMRI and must be mitigated as much as possible; in addition to motion correction and global signal regression, we performed censoring of high motion frames which has been shown to further mitigate these effects (Ciric et al., 2017) and included motion as a covariate in the GLM analysis."

Reviewer Comment

From the prior paper introducing the heritability estimates, roughly 70% of the FC variance was not due to heritable effects. However, the current paper reports that up to 78% of SC-FC coupling is estimated to be heritable. Some of this effect could be attributed to the region/system-level specificity discussed in the current paper. However, it could be helpful to more thoroughly discuss why SC-FC coupling might have higher heritability than FC in some cases (here again, having addressed any changes in estimates after preprocessing the data with more aggressive motion-correction procedures).

Response

Thank you for this important question regarding the heritability of the SC-FC coupling. In response to another comment, we recalculated the SC-FC coupling using Spearman rank correlation and recalculated SC-FC coupling heritability using this new estimate. We found that the heritability of SC-FC coupling is very similar to the heritability levels of FC and generally higher than the heritability levels of SC, see new Figure 5a in the manuscript (included below). See also our response to the related comment below for additional discussion of this point.

Reviewer Comment

Figure 5K shows a strong association between SC-FC coupling and FC heritability. The distribution of FC across regions likely averages to a higher heritability than that found in the 2017 study. Can this be quantified, and if so, why? Ruling out preprocessing and motion issues here also seems important. Further, if FC heritability drives (or is highly related to) SC-FC coupling, does this mitigate interest or relevance of SC-FC coupling as a unique measure? This question is related to my other theoretical and methodological points.

Response

After modifying our SC-FC coupling calculation to instead use Spearman rank correlation (in response to the comment below), we found that the heritability of SC-FC coupling has only a moderate correlation with the heritability of both FC and SC (see Figure 5k and l). In this way, SC-FC coupling heritability is not driven strongly by either FC or SC heritability, but instead is an independent measure of the heritability of the relationship between brain structure and function. We have made many revisions to the appropriate sections in the results and discussion to reflect these new findings.

Reviewer Comment

Like most connectomes, the degree and strength distributions of node-wise structural and functional connectivity are likely not perfectly Gaussian, and z-scoring alone will not remove influences like skewness and kurtosis of the distributions. These effects can strongly influence Pearson's correlation-based measures of SC-FC coupling. This should be addressed directly, for instance by applying a specific test of region-wise skewness and kurtosis of row-wise statistical distributions, then selecting the summary statistic accordingly or applying a transformation prior to subsequent analyses.

Response

Thank you for pointing out this issue, which we have resolved by using non-parametric Spearman rank correlation to calculate SC-FC coupling. When we used Spearman-rank to calculate the coupling, we also observed that the precision-based FC no longer resulted in higher coupling than the correlation-based FC matrices. This fact, in addition to our desire to match as closely as possible the Baum et al. (2020) paper for comparison purposes, meant that we now calculate all

Figure 5: SC-FC coupling heritability estimates. **a, d** and **g** Regional heritability estimates of SC-FC coupling, SC node strength and FC node strength. **b, e** and **h** Regional heritability estimates of SC-FC coupling, grouped by functional network, for SC-FC coupling, SC node strength and FC node strength, respectively. **c, f** and **i** Comparisons of heritability values between networks (t-statistics); those with FDR corrected $p > 0.05$ are marked with *ns*. **j** SC-FC coupling heritability has a weak, positive correlation with its signal strength (Pearson's $r = 0.124$, $p = 6.2e - 3$). **k** and **l** Regional heritability estimates of SC-FC coupling are significantly negatively correlated with regional heritability of SC node strength (Pearson's $r = -0.318$, $p \sim 0$) and significantly positively correlated with regional heritability of FC node strength (Pearson's $r = 0.311$, $p = 0$).

of our results based on the FC matrices derived using correlation (and not precision) based FC. Please see the extensive changes to the results, discussion and methods sections regarding the new analyses. Figure 2 in the manuscript is also included below.

Figure 2: Regional whole-brain SC-FC coupling varies spatially across the brain and is related to both within- and between-network coupling. **a** displays the SC-FC coupling for each cortical and subcortical region in the CC400 atlas. **b** shows the distribution of SC-FC coupling over regions grouped into nine different networks (the 7 cortical networks defined by Yeo et al. (Yeo et al., 2011), subcortical and cerebellum/brain stem). **c** shows the t-statistics for all pairwise comparisons of SC-FC coupling across networks, calculated as the network on the y-axis versus the network on the x-axis. Those comparisons with FDR corrected $p > 0.05$ are marked with *ns*. Visual and subcortical networks have higher SC-FC coupling than other networks while limbic and default mode areas have weaker SC-FC coupling than other networks. Abbreviations: VIS - visual, SOM - somatomotor, DATTN - dorsal attention, VATTN - ventral attention, LIM - limbic, FPN - frontoparietal, DMN - default mode, SUB - subcortical, CER/BS - cerebellum and brain stem. **d** Relationship between whole brain SC-FC coupling and the within-network SC-FC coupling (Pearson's $r = 0.416$, $p = 0$). **e** Relationship between whole brain SC-FC coupling and the between-network SC-FC coupling (Pearson's $r = 0.704$, $p = 0$). **f** Relationship between within- and between-network SC-FC coupling (Pearson's $r = 0.168$, $p = 8e - 4$).

We also have added some discussion of our findings that stand in contrast to the previous work comparing SC with precision-based FC and correlation-based FC, as we believe this is an important issue that has not yet been discussed in the literature. This paragraph has been added to the discussion:

"Functional activation flows not only through direct SC but also indirect, multi-synaptic white matter connections, which likely contributes to divergence of SC and FC to varying degrees (Suárez et al., 2020). Statistical, communication, biophysical and machine learning models have been applied to better align FC and SC (Abdelnour et al., 2014; Sanz Leon et al., 2013; Mišić et al., 2016;

Sarwar et al., 2020). Recent work has also found the strength of global SC-FC correlation depends on how FC is calculated (Liégeois et al., 0). In particular, that work showed FC calculated using partial correlation (precision), which aims to isolate direct and remove the effect of indirect functional connections, had stronger correlations with SC than standard FC calculated using full (Pearson) correlation. However, this observation was based on using Pearson correlation to assess global similarity of the upper triangular portions of the SC and FC matrices, which may not be an appropriate measure as SC is non-Gaussian. In fact, our analyses confirmed that using precision-based FC resulted in higher SC-FC coupling than correlation-based FC, but only when using Pearson correlation to measure SC-FC coupling. When using the more statistically appropriate Spearman correlation to assess the similarity of SC and FC, precision-based FC gives lower values (about half the magnitude) compared to correlation-based FC (see Supplementary Figure S9). We hypothesize this reduction in coupling may be driven by non-overlapping sparsity patterns that exist in both the SC and the precision-based FC."

Reviewer Comment

While the SC-FC measure used here is relatively simple, (row-wise Pearson's correlations between SC and FC elements), it is not the only way to encode this relationship. It emphasizes the connections between a node and all other regions, which does not appear to have a theoretical motivation in the current study. Other techniques such as alternatives to correlation (overlap indices, those that are more appropriate to non-Gaussian distributions), region-wise measures, or spectral decomposition (among others) could be evaluated. In addition, depending on the measure, associations with cognition and behavior can be strongly influenced (e.g., Medaglia et al., 2018). Why was this SC-FC technique used, other than for convenience? What is its theoretical motivation, and what alternatives could be justified?

Medaglia, J. D., Huang, W., Karuza, E. A., Kelkar, A., Thompson-Schill, S. L., Ribeiro, A., & Bassett, D. S. (2018). Functional alignment with anatomical networks is associated with cognitive flexibility. *Nature human behaviour*, 2(2), 156-164.

Response

Thank you for this very important comment. Two reasons for using correlation as a measure for SC-FC coupling are 1) simplicity of interpretation and 2) to directly compare to previous work done by Baum et al. (2020) in an adolescent population. To most closely match this previous publication, we have updated the SC-FC coupling to use Spearman correlation and changed the FC used in the analysis to be (full) correlation-based. We believe that our new results build upon and compliment the previous publication and, in doing so, furthers our understanding of structure-function coupling in young adult populations. We have added the following sentences to the Methods section:

"We chose non-parametric Spearman-rank correlation to quantify the similarity of a region's structural and functional connectivity pattern to the rest of the brain as it is a measure that is straightforward and easily interpreted and, importantly, accommodates the non-Gaussianity of the entries in the SC. In addition, we wanted to compare the results found here in young adults to previous work using a similar approach in adolescents wherein Spearman-rank correlation was used to quantify SC-FC alignment (Baum et al., 2020)."

We believe whole-brain analysis provides a valid measure of alignment that is distinct from within and between-network alignment. The whole-brain SC-FC coupling measure allows straightforward comparison of the various regions in the brain to one another and does not rely on functional system definitions that are not fixed and can vary depending on the atlas/method of definition particularly in higher-order areas. There are also statistical concerns about the varying number of regions in the atlas assigned to each network that may contribute to different signal-to-noise ratio values in the coupling calculation across regions. Considering these potential limitations and confounds, and our desire to match as closely as possible previous work in adolescents, we decided to present the main results of whole-brain SC-FC coupling. However, we have added some comparison of the whole-brain coupling to within and between-system coupling to the main paper and report full results in the supplemental material (see the next comment and response for details).

Reviewer Comment

Further, the analyses do not encode system-wide constraints when computing the SC-FC measures (for instance, only including system-to-system elements in the analysis). This muddies some of the analyses because region-wise and system-wise effects are not well-disentangled. The system-level findings appear to involve nodes with projections to every system, not just system-to-system connections. Further addressing this issue seems justified.

Response

Thank you for this comment. The emphasis of this work is to study whole brain alignment of SC and FC, not to investigate between and within-system alignment separately as described in your comment. We believe using a whole-brain approach provides interesting, novel findings and allows straightforward, easy to interpret results. In addition, this work aims to extend findings reported on adolescent populations using a similar whole-cortex coupling analysis. However, we do think comparing the within and between network SC-FC coupling to our whole-brain SC-FC coupling may provide insight as to what it reflects. Therefore, we calculated the within and between-system SC-FC coupling separately using the 9 functional network assignments (7 Yeo plus subcortical and cerebellar/brain stem) of each of the 392 regions (see Figure S2a and d, which are now included in the manuscript as Supplemental Information Figure S2). We added this text to the Methods:

"To assess the association of between and within-network coupling to whole-brain coupling, we separately calculated, for each region, its between and within-network SC-FC coupling as follows. Within-network SC-FC coupling for each region was the Spearman correlation of the structural and functional connections between that region and other regions in the same network; between-network SC-FC coupling was the same calculation but between that region and regions outside of its assigned network. To compare these two network-specific measures to whole brain SC-FC coupling, we calculated Pearson correlation between the measures."

and this text to the Results:

"When comparing whole-brain SC-FC coupling to the within and between-network coupling, we found that, unsurprisingly, whole brain coupling was highly correlated with the between-network SC-FC coupling (Pearson's $r = 0.704$, $p = 0$) and moderately correlated with the within-network coupling (Pearson's $r = 0.416$, $p = 0$). Within network coupling was higher overall than between network coupling; within-network coupling was particularly high within certain visual regions (see Supplementary Information Figure S2). Regions in the ventral attention network had the most

disparate within and between-network coupling strengths, where it had significantly lower within-network coupling than all other networks and significantly higher between-network coupling than 5 of the other 8 networks (see Supplementary Figure S2). Finally, we observed that SC-FC coupling was also moderately positively correlated with SC node degree (Pearson's $r = 0.281$, $p = 0$) but not correlated with FC node degree (see Supplementary Figure S3)."

and this text to the Discussion:

"Finally, whole-brain SC-FC coupling appeared to be more driven by between network coupling than within network coupling. This is likely because of the larger overlap in regions included in the between-network calculation. One issue with calculating the within- and between-network coupling is that the number of regions in the CC400 atlas assigned to each of the 7 Yeo networks is not equal (range: 22 – 79). Thus, the within and between-network coupling is biased and likely noisy for networks that have a smaller number of regions than ones with a larger number of regions which complicates comparison."

Figure S2: Within- and between-network SC-FC coupling. **a** Within-network SC-FC coupling for each region is the Spearman correlation of the structural and functional connections between that region and other regions in the same network. **b** Within-network SC-FC coupling in nine different networks. **c** Pair-wise comparisons of the within network coupling. **d** Between-network SC-FC coupling for each region is the Spearman correlation of the structural and functional connections between that region and other regions outside of its assigned network. **e** Between network SC-FC coupling in nine different networks. **f** Pair-wise comparisons of the between network coupling values. Dorsal/ventral attention and subcortical areas have significantly higher between-network coupling than other networks while cerebellum and brain stem have significantly lower between-network coupling than other networks.

Reviewer Comment

Informalisms on line 269, 271, section starting at 278: the colloquial term “fingerprint” has been popularized, but the brain is not a finger and structural and

functional connectivity (and their association) are not prints. Language describing the exact methods and their interpretation would be clearer. For instance, at line 268, the authors state that “coupling was constructed by calculating the Pearson correlation between a row of the SC matrix, representing the connectivity fingerprint of that region in the brain”. Omitting the phrase “representing the connectivity fingerprint of that region in the brain” results in a clear statement. In addition, it reduces the impression that the only way to represent structure-function coupling is using row-wise Pearson’s correlations between structural and functional data.

Response

We agree with your assessment of this imprecise wording; we have deleted that part of the sentence and removed all references to the term "fingerprints".

Reviewer Comment

Atlas-based analyses often overlook important consequences on the neural data and the corresponding interpretation. For instance, the microanatomy, cell body density, and anatomical connections involving the basal ganglia and cerebellum are quite different from the cortex. How should findings concerning those regions be couched and what appropriate future directions could be pursued?

Response

While we agree there are differences in microanatomy and connection type for subcortical, cerebellar and brain stem regions, we also believe that totally excluding them from the analyses (as previous work has at times done) ignores potentially important information. Subcortical and cerebellar regions certainly play a large role in the brain’s functional activation patterns, and regions like the thalamus are very highly structurally connected to much of the cortex. We believe that ignoring these regions in any analysis of structure and function would result in an incomplete picture of the brain-wide SC-FC coupling and may potentially overlook important findings. We do agree that some of the regions considered and connections measured may have varying underlying anatomical and/or functional properties, and we have added the following discussion to the limitations section:

"Furthermore, in this whole-brain, atlas-based analysis of SC-FC coupling, all connections and regions are treated identically, even those in the cerebellum, subcortex and brainstem. We believe that these regions play a very important role in overall patterns of brain activity and white matter connections so we included them here; however, we also acknowledge that their microanatomy and anatomical connection type (inhibitory vs excitatory) may differ from that of cortical regions. Future work may attempt to modify the SC-FC coupling measure to account for these differences, e.g. treating inhibitory connections differently from excitatory connections."

Response To Reviewer #2

Overall Comments

In “Regional structural-functional connectome coupling is heritable and associated with age, sex and cognition in adults” Gu et al explore the distribution, reliability, and heritability of regional structure-function coupling in young adults, and examine how it relates to age, sex, and cognition. This paper has a great deal to recommend it, including an interesting and important topic, appropriate use of a large data resource like the HCP, clear writing, excellent visualizations, code availability, and appropriate methods. Overall, I think it makes a valuable contribution and will be of interest to a broad readership. That being said, I have several comments for the authors to consider; I have signed this review for transparency and am happy to discuss these comments with the authors if they are unclear. -Ted Satterthwaite

Response

We would like to thank you for your positive feedback. Your detailed comments have considerably helped to improve the quality of the revised manuscript.

Reviewer Comment

Sample selection. For readers who are not intimately familiar with the HCP, the methods describing the sample selection criteria for the various analyses are somewhat opaque. Inclusion of a flowchart for inclusion with clear specification of QA criteria for all modalities used would be informative.

Response

Thank you for your suggestion; this is indeed an important point when analyzing HCP data that is often overlooked when describing the data. We now include this flowchart figure (copied here below) in the Supplemental Information (Figure S1) to fully specify how we identified the appropriate samples for each analysis.

Reviewer Comment

Regularization of FC matrices. The methods for generating the precision matrices was somewhat hard to follow; slowly stepping through these methods (potentially with a schematic) would make this more accessible. For example, how was overfitting avoided against in the grid search / parameter selection process? Was cross-validation used?

Figure S1: Flowchart illustrating the selection of the HCP data for each of the analyses. We began with the final S1200 HCP release, of which only 941 subjects had all four resting-state functional and diffusion MRI scans. We used these 941 in the heritability analyses in the main paper and 645 of these subjects that were white and non-Hispanic in the subgroup analysis included in Figure S5. A set of 41 of the 941 had another visit 6 months after the initial visit, which comprised the group of individuals in the test-retest analysis. An unrelated subset of 420 out of the 941 were randomly chosen for the calculation of the SC-FC coupling; 415 out of this set of 420 had composite cognition scores and were included in the GLM analysis. A second set of 346 unrelated individuals (non-overlapping with the 420) was selected for the out-of-sample validation study.

Response

In the original version of the manuscript, precision-based FC was calculated exactly as in (Liégeois, Raphaël, et al. "Revisiting correlation-based functional connectivity and its relationship with structural connectivity." *Network Neuroscience* (2020): 1-17.) However, in the process of responding to R1's comments, we have updated the main results in the manuscript to use full correlation-based FC instead of precision-based FC. We do think that our new findings on precision-based FC versus full correlation-based FC are important in the context of past and future work looking at SC-FC coupling, see Figure S9 and the text in the discussion, copied here for convenience.

"However, this observation was based on using Pearson correlation to assess global similarity of the upper triangular portions of the SC and FC matrices, which may not be an appropriate measure as SC is non-Gaussian. In fact, our analyses confirmed that using precision-based FC resulted in higher SC-FC coupling than correlation-based FC, but only when using Pearson correlation to measure SC-FC coupling. When using the more statistically appropriate Spearman correlation to assess the similarity of SC and FC, precision-based FC gives lower values (about half the magnitude) compared to correlation-based FC (see Supplementary Figure S9). We hypothesize this reduction in coupling may be driven by non-overlapping sparsity patterns that exist in both the SC and the precision-based FC."

Reviewer Comment

Figure S9: SC-FC coupling calculated using precision-based FC. a SC-FC coupling across cortical, subcortical and cerebellar regions. **b** SC-FC coupling distribution across nine networks. Ventral/dorsal attention, frontal parietal and somatomotor networks had generally higher coupling than other areas. Limbic and subcortical area had weaker mean coupling (0.08 ± 0.03 and 0.08 ± 0.02 , respectively). **c** Pair-wise comparisons between network coupling. Pearson's correlation of the SC-FC coupling results in the main paper (using full correlation-based FC) with the precision-based FC measure is $r = 0.486$ ($p = 0$).

Test/retest section. I thought this was a very valuable element of the paper, and as such was surprised to see that there was (as far as I can tell) zero description of these procedures in the methods. While this oversight can be easily remedied, it would be additionally quite informative to expand this section and contrast SC/FC coupling with the component measures, like the analyses included in the heritability analyses – i.e., SC, FC matrices. Furthermore, it would be useful to include supplementary analyses evaluating the relative reliability of precision-FC vs. correlation-FC (for both FC itself and SC/FC coupling). This would also potentially help contextualize the heritability analyses – i.e., was low heritability of SC due simply to noise in the SC measures? Does precision FC allow for greater reliability?

Response

Thank you for bringing our attention to this issue. We have now added a more detailed description in the methods section regarding the reliability analysis, including the data description and the measures used to assess reliability in Methods (see below). We also calculated the reliability estimates for SC and FC node strength (the measures used in the heritability analysis) and it does appear that FC and SC degree are quite reliable across time and different populations, so we do not think that differences in noise of the measures contributed to the varying heritability values. Since we do not use precision-based FC in the main part of the new version of the manuscript, we did not show the reliability of this measure.

"Forty-one subjects in HCP had a second MRI scan approximately six months after the first scan (test-retest). The replication (out-of-sample) analysis used another subset of 346 unrelated HCP subjects (age, 28.78 ± 3.80 y; 148 males and 198 females), distinct from the initial set of 420 unrelated subjects. It should be noted that, while each set of subjects did not contain relatives within them, there may be some familial relationships across the two sets of subjects which could result in an overestimation of the out-of-sample reliability."

and

"Reliability of SC-FC coupling, SC node strength and FC node strength was assessed by calculating Pearson correlation between the three measures extracted from the test and retest visits ($N = 41$) and between the measures extracted from the original sample ($N = 420$) the out-of-sample population ($N = 346$). Bland-Altman plots were also used to quantify the reliability of SC node strength, FC node strength and SC-FC coupling, which gave us level of agreement (LoA) for each of the measures. The mean difference, also called the bias, is calculated by $\bar{d} = \frac{1}{n} \sum_{i=1}^n d_i$ and the LoA between the test-retest and out-of-sample replication studies are defined by a 95% prediction interval of a particular value of the difference which are computed as $\bar{d} \pm 1.96S_d$ where $S_d = \sqrt{\frac{1}{n-1} \sum_{i=1}^n (d_i - \bar{d})^2}$."

and we also mention the reliability of FC/SC node strength in the results:

"Next, we tested the reliability and reproducibility of SC-FC coupling by examining its consistency within individuals over time and across different populations of individuals. To test for consistency over time within the same individuals, we used data from a subset of 41 subjects who had a second MRI 6 months after the first. SC-FC coupling was indeed highly consistent across this time period, with a mean difference of $\mu = -0.002$, limits of agreement $LoA = \mu \pm 0.034$, see Figure 3a, and a test-retest correlation of 0.977 ($p = 1.397e - 264$). Furthermore, we examined out-of-sample, across population reliability in SC-FC coupling using a subset of 346 unrelated HCP subjects (age, 28.78 ± 3.80 y; 148 males and 198 females), distinct from the initial set of 420 unrelated subjects. Out-of-sample reliability was also high, with a small mean difference $\mu = 0.005$

and limits of agreement $LoA = \mu \pm 0.012$, see Figure 3b, and high correlation (Pearson's $r = 0.997$, $p = 0$). Reliability of SC node degree and FC node degree was also very high, with a test-retest and out-of-sample correlation of $r = 0.995$, $p = 0$ and $r = 0.999$, $p = 0$ for FC node degree and $r = 0.998$, $p = 0$ and $r = 0.999$, $p = 0$ for SC degree, respectively, see Supplementary Figure S4."

and discussion:

"From the reliability analysis, it does not appear that the SC's lower heritability values are due to increased measurement noise, as SC node strength was as reliable as FC and SC-FC coupling."

Reviewer Comment

Distance dependence. Tractography methods have a known distance dependence. While the appropriate preprocessing used may minimize this, additional analyses where inter-node Euclidean distance is treated as a covariate would bolster confidence that the observed results were not due to this bias.

Response

Thank you for this suggestion; we agree that there can be biases in the tractography results due to distance. In fact, this is a reason we chose to perform global filtering using SIFT2, which aims to adjust this distance bias. Because we used this approach, we think that the remaining SC strengths are not so biased by distance. Still, for comparison, we did perform this analysis but instead include the results in the Supplemental Information (Figure S10, see below).

We have added the following text to the results:

"Biases in tractography algorithms exist, including the effect of distance between regions which we adjusted for somewhat using a global filtering approach (Smith et al., 2015). SC-FC coupling calculated using partial Spearman-rank with distance between pairs of regions' centroids as a covariate show similarities with the main coupling results (Supplementary Figure S10). One noticeable difference between the two coupling calculations was weaker subcortical SC-FC coupling when distance was considered in the calculation. We hypothesize this is due to the fact that subcortical structures are further from the majority of cortical regions but also highly connected to all of them so covarying for distance has a greater impact on its coupling measures."

Reviewer Comment

Results in discussion. There are several instances of new results being presented in the discussion section. These would be better placed in the results section, with transparent procedures reported in the methods.

Response

Thank you for bringing our attention to this issue. We have addressed these outstanding sentences in the new version of the manuscript.

a Partial SC-FC coupling (distance)

Figure S10: Partial SC-FC coupling with inter-node Euclidean distance as a covariate. **a** Partial SC-FC coupling was computed by partial Spearman-rank correlation of the row in SC and its corresponding row in FC with the Euclidean distance between two regions as a covariate. It varies across cortical and subcortical areas and ranges from -0.03 to 0.39 . **b** Partial SC-FC coupling in nine networks. **c** Pair-wise comparisons between nine networks. Visual and somatomotor network have significant higher partial SC-FC coupling than other networks. Limbic network has significantly weaker partial SC-FC coupling. Partial SC-FC coupling is correlated with the standard SC-FC coupling (Pearson's $r = 0.431$, $p = 0$).

Reviewer Comment

Comparing regional profiles. There are multiple figures/analyses in the paper where regional profiles are compared – for example, figure 5 comparing heritability of SC to coupling. R values are provided to estimate effect size, and p values seem to be based on parametric statistics. While the analyses are useful, the p values from significance tests are not valid due to the non-independence of measures and the arbitrary atlas resolution; i.e., greater “statistical power” could be achieved by simply using a higher resolution atlas (i.e., 100 nodes vs. 10,000 nodes). Testing procedures that consider these factors should be used—i.e, the “spin test” (<https://www.sciencedirect.com/science/article/abs/pii/S1053811918304968>) or “brains smash” (<https://brainsmash.readthedocs.io/en/latest/>). Given the robustness of the findings, I do not imagine this will substantially alter the results.

Response

Thank you for this suggestion. For all the network-wise group comparisons and correlations of regional measures, we now calculate p -values using a permutation test with 10000 resamples. See new text in the methods:

"To compare these two network-specific measures to whole brain SC-FC coupling, we calculated Pearson correlation between the measures; p -values were calculated using a permutation test with 10000 resamples."

"Pairwise comparisons were done within the networks and the heatmaps in each figure show the unpaired t-statistic comparing the network level values. Significance of the t-statistic was quantified using a permutation test with 10000 random resamples. This was done to avoid bias introduced via the number of atlas regions in each network."

Response To Reviewer #3

Overall Comments

In the current paper Gu and colleagues studied SC-FC coupling, the association between structural and functional connectivity, and its genetic basis, and association with age, sex, and cognitive abilities in the HCP sample. As reported in previous work (Vázquez-Rodríguez, 2019; Baum, 2020), the researchers observed SC-FC coupling was highest in visual and somatomotor areas, and lower in association regions, sub-cortex and cerebellum. Next, the authors show that SC-FC coupling decreases with age, and is associated with sex and cognition. Last, they observed that the coupling of SC-FC is heritable, especially in visual, dorsal attention, and fronto-parietal networks, more than SC or FC alone.

The current insights in structure-function coupling are largely in line with previous papers on this topic such as Vázquez-Rodríguez, 2019 or Baum, 2020. Although the associations with sex and its heritability are interesting, I am not very sure whether the insights gained in the current study extend previous knowledge significantly. I hope my comments can be of help.

Response

We would like to thank you for your feedback. Your detailed comments have considerably helped with improving the manuscript. We believe that changes made in response to your and other reviewer's comments have greatly improved the manuscript.

Reviewer Comment

Why did the authors perform global signal regression? Did results remain similar when not running GSR?

Response

We chose to use GSR as it can help to remove non-neuronal BOLD effects such motion and respiration; however we do agree that a comparison between the methods is warranted. We now show that the SC-FC coupling calculated using FC without GSR are highly correlated (Pearson $r = 0.961$, $p = 0$). We include the figure below in the new supplemental information, and discuss it in the Sensitivity Analysis subsection in the Results:

"We also see good agreement with the main SC-FC coupling values when using FC derived 1) without global signal regression (see Supplementary Figure S8) and 2) using partial correlation (precision) (Supplementary Figure S9)."

Reviewer Comment

a SC-FC (without GSR) coupling

b

c

Figure S8: SC-FC coupling computed using FC without global signal regression (GSR). **a** SC-FC (without GSR) coupling varies across cortical and subcortical areas with range from -0.01 to 0.39. **b** SC-FC (without GSR) coupling in nine networks. **c** Pair-wise comparisons between nine networks. Pearson's correlation of the SC-FC coupling results in the main paper (with GSR) with the non-GSR coupling was $r = 0.961$ ($p = 0$).

Did the authors consider the lack of cross hemisphere connections using DTI?

Response

This is indeed a point to consider, we have added many secondary analysis to assess the robustness of the SC-FC coupling, including one to address this comment. We calculated the within hemisphere SC-FC coupling (see Figure 8) and found it to be overall highly consistent with the whole-brain SC-FC coupling (Pearson's $r = 0.864$ ($p = 0$)).

Figure S11: Within hemisphere SC-FC coupling. **a** SC-FC within hemisphere coupling varies across cortical and subcortical areas with range from -0.02 to 0.52, which is a bit higher than whole brain SC-FC coupling but preserves consistency with the whole-brain results. **b** Within hemisphere SC-FC coupling in nine networks. **c** Pair-wise comparisons between nine networks. Pearson's correlation of the SC-FC coupling results in the main paper (using whole-brain SC/FC) with the single hemisphere SC/FC coupling is $r = 0.864$ ($p = 0$).

We added the following to the "Sensitivity analyses" subsection in the Results:

"It is also known that tractography algorithms underestimate cross-hemisphere connections; SC-FC coupling within a single hemisphere was very similar to whole-brain SC-FC coupling (Supplementary Figure S11), indicating minimal effects of the under-estimated inter-hemispheric SC on the coupling calculations."

Reviewer Comment

Why did the authors use the CC400 atlas? Would an anatomically defined or functionally informed atlas not be more appropriate when assessing structure-function relationships?

Response

We chose the functionally-defined CC400 atlas since it is fine-grained and has cortical, subcortical and cerebellar regions. However, we do agree it is interesting to validate the main results with an anatomically-defined atlas. Therefore, we re-analyzed the data using a 191 region cortical, subcortical and cerebellar atlas from FreeSurfer (Destrieux et al., 2010) in Figure S6.

Figure S6: SC-FC coupling in FS191 atlas. **a** SC-FC coupling in FS191 atlas varies across cortical and subcortical areas with range -0.02 to 0.40 . **b** SC-FC coupling distribution in nine networks. Visual, frontal parietal network, cerebellum/brain stem had generally higher coupling than other areas, with mean coupling 0.23 ± 0.06 , 0.24 ± 0.06 and 0.24 ± 0.06 , respectively. **c** Pair-wise comparisons between networks.

The following text was included in the results section:

"First, we recalculated SC-FC coupling using anatomically-derived 191 region atlas from FreeSurfer (Destrieux et al., 2010) (Supplementary Figure S6); the coupling values appear very similar to the main SC-FC results as do the results of the GLM analyses (Supplementary Figure S7). "

Reviewer Comment

For the association with age, is the sample's age range really appropriate to make statements about ageing? What would aging mean in a young adult sample? Why not use a second sample with a broader age-range for such an evaluation?

Response

Previous work has been done in developing, adolescent populations and we aimed to replicate their findings in our young adult population (22 – 37 years), a time frame where development in the brain is still occurring. Our new findings (changed in response to comments from other reviewers) indicate less of an association between SC-FC coupling and age than the original version. The regions that did have significant effects in our young adult study agreed with those identified as being associated with age in an adolescent population (Baum et al., 2020), which lends credence our findings. We have modified the discussion to state:

"Despite the limited age range of our sample (22 – 37 years) we still observed a few associations between SC-FC coupling and age, with stronger medial orbito-frontal SC-FC coupling and weaker cerebellar coupling being related to increased age. Processes like synaptic pruning, functional diversification and myelination that may impact SC-FC coupling, and are classically associated with adolescent populations, are still occurring in young adults through at least the mid-20s. Orbitofrontal regions of the prefrontal cortex, particularly important in impulse control, are among the last regions in the brain to fully develop (Giedd et al., 1999; Torregrossa et al., 2008). Interestingly, Baum et al. (2020) found mostly age-related increases (including in medial orbitofrontal regions in agreement with our current findings) and some decreases in SC-FC coupling with increased age during adolescence. Their age-related associations were indeed much more widespread than our findings in young adults, indicating, unsurprisingly, more dynamic SC-FC coupling in adolescence that continues in some prefrontal regions into young adulthood."

Reviewer Comment

It is not clear to me what the network-based coupling is often inverse (Figure 2C) between the association from network $X > Y$ versus $Y > X$.

Response

Thank you for highlighting this potentially confusing part of the figures. In the original version, we were showing the t-statistic for the network on the vertical axis minus the network on the horizontal axis, which resulted in a matrix that was symmetric except for a sign flip. We agree this was not an optimal visualization and have greyed out the upper triangular portion of the matrix for increased clarity of the results.

Reviewer Comment

Did the authors control for ICV in their analysis of the relationship between sex and/or cognition and SC-FC coupling?

Response

Thank you for this detailed question. Yes, we included ICV as a covariate, please see the Methods section.

Reviewer Comment

For the heritability analyses, it seems that the FC / SC strength is a new and different measure, and I am unsure how this would relate to SC-FC coupling. As such comparing the heritability of FC to SC-FC coupling is unclear to me. Looking at the patterns, aren't heritability values a bit biased by the signal strength? E.g. subcortical regions and limbic regions have much lower h2r and signal strength?

Response

We compared the heritability of SC-FC coupling with the heritability of SC and FC in order to show that it is a new phenotype that does not depend strongly on the underlying heritability of either SC or FC alone. After modifying our coupling calculation in response to reviewer 1's comments, we found that SC-FC coupling heritability is only moderately correlated with the heritability of SC and FC in opposing directions (see Figure 5k and l).

We agree that it is possible heritability estimates are influenced by the magnitude of the phenotype. To test this, we calculated the correlation between SC-FC coupling and their corresponding heritability estimates. We found a small positive correlation (see Figure 5j attached below, Pearson $r = 0.124$, $p = 6.2e - 3$), indicating that generally SC-FC coupling heritability is not determined by its magnitude. This panel is included in the new manuscript in Figure 5j. In fact, one important counter-example in the new version's analysis using Spearman-rank SC-FC coupling shows subcortical structures actually have very high SC-FC coupling magnitude but their heritability is lower than most other networks.

We added the following text to the results section:

"SC-FC coupling strength was weakly correlated with its heritability (Pearson's $r = 0.124$, $p = 6.2e - 3$, see Figure 5j), suggesting that SC-FC coupling heritability is not driven by its magnitude."

Reviewer Comment

Possibly it would help to add standard deviations, measures of variance to the mean values reported in the text.

Figure 5j: Relationship if SC-FC coupling signal strength and its heritability.

Response

Thank you for the suggestion. We have added the standard deviations associated with the mean values in the text.

Reviewer Comment

Why is the association with average myelination discussed in the discussion but not in the results?

Response

We have removed this part of the discussion, as we agree it should not have been in that section.

Reviewer Comment

In the discussion it was mentioned the heritability of structural connectivity was presented for the first time. However, this has been reported before in different contexts (such as heritability of T1wT2w ratio or DTI metrics)

Response

Thank you for pointing us to this missing literature regarding the heritability of white matter measures. We have added the related papers to the introduction:

"Several recent publications have revealed the varying degrees to which the brain's FC (Ge et al., 2017; Sinclair et al., 2015; Miranda-Dominguez et al., 2018) and white matter microstructure, measured with diffusion MRI summary statistics like fractional anisotropy and mean diffusivity, are heritable (Vuoksimaa et al., 2017; Zhao et al., 2019). Very few studies explore heritability of SC networks; however, some recent preliminary work investigated the relationships between gene co-expression, single nucleotide polymorphisms (SNPs), FC, and SC in a developmental cohort (Bertolero et al., 2019). In particular, this recent work suggests that gene coexpression and SNPs are consistently more strongly related to FC than to SC, and furthermore, that the brain's FC architecture is potentially the mediating factor between genetic variance and cognitive variance across the developing population. "

and we have changed the discussion to state:

"Interestingly, we found highest SC heritability in limbic and subcortical networks, which were the networks with the lowest heritability in FC and SC-FC coupling. Previous work has suggested different genetic signatures underlying brain anatomy and physiology (Glahn et al., 2010). However, these areas do tend to have the most noise in fMRI which could also contribute to lower FC heritability estimates. While no other studies have investigated the heritability of SC, one recent study quantifying heritability of the size of cortical areas (as defined by FC) showed unimodal motor/sensory networks had higher heritability (0.44) relative to heteromodal association networks (0.33) Anderson et al. (2021). We do show mixed agreement with their findings in that unimodal visual networks, but not somato-motor networks, had highest anatomical SC heritability across cortical networks. "

Response To Reviewer #4

Overall Comments

In this manuscript, Zijin Gu and colleagues describe their investigation of regional structural-functional connectome (SC-FC) coupling in a sample of healthy young adults aged 22 to 37 from the Human Connectome Project (HCP) cohort. They investigate sex and age differences, the relationship with cognitive ability, and estimate heritability. The study is novel and fills a gap in knowledge in the sense that it investigates regional SC-FC coupling in a large MRI cohort. Overall, the manuscript is well written and easy to follow, and it is of interest to the broader field of human neuroscience. In my view, the methods are appropriate and well executed. I have a few suggestions/comments for the authors that could help make the paper stronger.

Response

We would like to thank you for your positive feedback. Your detailed comments have considerably helped with improving the quality of the revised manuscript.

Reviewer Comment

I feel that the paper falls short in terms of the interpretation and providing biological insights. The authors could expand their discussion of findings in the context of the broader SC and FC literature.

Response

Thank you for this suggestion on expanding the discussion in terms of interpretation and biological insights. We have expanded the discussion section in the new version of the manuscript.

Reviewer Comment

Heritability is a population-specific parameter. The HCP cohort is ethnically diverse. The number of white non-Hispanic individuals is $N = 720$. I would suggest running a sensitivity analysis using only this homogeneous subset of individuals.

Response

Thank you for drawing our attention to this point. We re-ran the SC-FC coupling heritability analysis on only the white and non-Hispanic individuals ($N = 645$, see Figure 11a) and found very similar results to the SC-FC coupling heritability of all individuals (Pearson's $r = 0.972$, $p = 0$, see Figure S12b). We added this figure to the supplementary material and added text to the Methods:

Figure S12: Heritability of SC-FC coupling for only white and non-Hispanic individuals. **a** Heritability of the white and non-Hispanic subgroup ranges from 0 to 0.64 and varies across cortical and subcortical regions. **b** The subgroup heritability is highly consistent with the heritability of SC-FC coupling for all HCP individuals (Pearson's $r = 0.972$, $p = 0$).

"Finally, because there may be differences in genetic similarity patterns across race/ethnicity, we re-calculated heritability of the various measures using a homogeneous sub-set of white, non-Hispanic individuals ($N = 645$)."

and Results section of the paper:

"Finally, we observe that the varied race/ethnicity of the 941 individuals does not have much influence on heritability estimates; a subgroup analysis of white, non-Hispanic individuals revealed very similar heritability patterns in SC-FC coupling (Pearson's $r = 0.972$, $p = 0$), see Supplementary Figure S12."

Reviewer Comment

The authors could try to disentangle the relationship between regional SC-FC coupling and crystallized vs. fluid intelligence separately.

Response

Thank you for the suggestion. Our motivation was not to study specific aspects of cognitive function, but overall cognitive scores which is why we selected the total composite cognitive score. We did perform the analysis using crystallized and fluid intelligence scores no significant relationships were identified in the GLM analysis. We reasoned that these two measures may have lower SNR compared to the total composite cognitive scores. We did add this sentence to the Limitations and Future work section:

"Measuring cognition is not an easy task; we chose here to investigate the highest-level composite score (total cognition) but future work could explore more specific cognitive scores like crystallized and fluid intelligence."

Reviewer Comment

The authors could include years of education in the GLM (e.g., to explore whether the association between SC-FC coupling is somewhat mediated/moderated by education)

Response

Thank you for the suggestion, this is indeed an important consideration that we originally overlooked. We have added years of education as a covariate to the GLM analysis, as well as the interaction between education and total composite cognition scores. No significant relationships were found in the GLM analysis with SC-FC coupling and education or the education*cognitive score interaction term included in the GLM. See this new text in the Methods section:

"There are several different covariates that we hypothesized may have significant relationships with SC-FC coupling, namely, age, sex, years of education, total cognitive score, intracranial volume (ICV) and in-scanner head motion...Finally, using a generalized linear model (GLM) approach, we assessed regional associations between SC-FC coupling and in-scanner motion, demographics and cognitive scores, plus four interaction terms (age*cognitive score, sex*cognitive score, years education*cognitive score and ICV*motion)."

and this text in the results:

"There were a mix of positive and negative associations found between SC-FC coupling and in-scanner head motion (see Supplementary Figure S5); no other covariates in the GLM model had significant relationships with SC-FC coupling."

Reviewer Comment

Line 18 - I would suggest changing the term "pathological populations".

Response

We have changed that term to "disordered populations", which we agree is more appropriate.

Reviewer Comment

In lines 45-50, the authors talk about heritability and then genetic co-expression. This is a bit confusing.

Response

We have modified the text to state:

"Most studies have focused on FC; however, some recent preliminary work investigated the relationships between gene co-expression, single nucleotide polymorphisms (SNPs), FC, and SC in a developmental cohort Bertolero et al. (2019). In particular, this recent work suggests that gene co-expression and SNPs are consistently more strongly related to FC than to SC, and furthermore, that the brain's FC architecture is potentially the mediating factor between genetic variance and cognitive variance across the developing population."

Reviewer Comment

Figure 3 could be supplementary.

Response

We believe that test-retest reproducibility and multi-sample validation should be a prominent feature of scientific papers, so we have chosen to keep this figure in the main text.

Reviewer Comment

In the legend of Figure 5, there is a typo: "Rregional"

Response

Thank you for pointing out this type, we have corrected it.

References

- Abdelnour, F., Voss, H. U., and Raj, A. (2014). Network diffusion accurately models the relationship between structural and functional brain connectivity networks. *NeuroImage*, 90:335–47.
- Anderson, K. M., Ge, T., Kong, R., Patrick, L. M., Spreng, R. N., Sabuncu, M. R., Yeo, B. T., and Holmes, A. J. (2021). Heritability of individualized cortical network topography. *Proceedings of the National Academy of Sciences*, 118(9).
- Baum, G. L., Cui, Z., Roalf, D. R., Ciric, R., Betzel, R. F., Larsen, B., Cieslak, M., Cook, P. A., Xia, C. H., Moore, T. M., Ruparel, K., Oathes, D. J., Alexander-Bloch, A. F., Shinohara, R. T., Raznahan, A., Gur, R. E., Gur, R. C., Bassett, D. S., and Satterthwaite, T. D. (2020). Development of structure–function coupling in human brain networks during youth. *Proceedings of the National Academy of Sciences*, 117(1):771–778.
- Bertolero, M. A., Blevins, A. S., Baum, G. L., Gur, R. C., Gur, R. E., Roalf, D. R., Satterthwaite, T. D., and Bassett, D. S. (2019). The network architecture of the human brain is modularly encoded in the genome. *arXiv*.
- Ciric, R., Wolf, D. H., Power, J. D., Roalf, D. R., Baum, G. L., Ruparel, K., Shinohara, R. T., Elliott, M. A., Eickhoff, S. B., Davatzikos, C., Gur, R. C., Gur, R. E., Bassett, D. S., and Satterthwaite, T. D. (2017). Benchmarking of participant-level confound regression strategies for the control of motion artifact in studies of functional connectivity. *NeuroImage*, 154:174–187.
- Destrieux, C., Fischl, B., Dale, A., and Halgren, E. (2010). Automatic parcellation of human cortical gyri and sulci using standard anatomical nomenclature. *Neuroimage*, 53(1):1–15.
- Ge, T., Holmes, A. J., Buckner, R. L., Smoller, J. W., and Sabuncu, M. R. (2017). Heritability analysis with repeat measurements and its application to resting-state functional connectivity. *Proceedings of the National Academy of Sciences*, 114(21):5521–5526.
- Giedd, J. N., Blumenthal, J., Jeffries, N. O., Castellanos, F. X., Liu, H., Zijdenbos, A., Paus, T., Evans, A. C., and Rapoport, J. L. (1999). Brain development during childhood and adolescence: a longitudinal mri study. *Nature neuroscience*, 2(10):861–863.
- Glahn, D. C., Winkler, A. M., Kochunov, P., Almasy, L., Duggirala, R., Carless, M. A., Curran, J. C., Olvera, R. L., Laird, A. R., Smith, S. M., Beckmann, C. F., Fox, P. T., and Blangero, J. (2010). Genetic control over the resting brain. *Proceedings of the National Academy of Sciences*, 107(3):1223–1228.
- Liégeois, R., Santos, A., Matta, V., Van De Ville, D., and Sayed, A. H. (0). Revisiting correlation-based functional connectivity and its relationship with structural connectivity. *Network Neuroscience*, 0(ja):1–25.
- Miranda-Dominguez, O., Feczko, E., Grayson, D. S., Walum, H., Nigg, J. T., and Fair, D. A. (2018). Heritability of the human connectome: A connectotyping study. *Network Neuroscience*, 2(2):175.
- Mišić, B., Betzel, R. F., de Reus, M. A., van den Heuvel, M. P., Berman, M. G., McIntosh, A. R., and Sporns, O. (2016). Network-Level Structure-Function Relationships in Human Neocortex. *Cerebral cortex (New York, N.Y. : 1991)*, 26(7):3285–3296.

- Sanz Leon, P., Knock, S., Woodman, M., Domide, L., Mersmann, J., McIntosh, A., and Jirsa, V. (2013). The virtual brain: a simulator of primate brain network dynamics. *Frontiers in Neuroinformatics*, 7:10.
- Sarwar, T., Tian, Y., Yeo, B., Ramamohanarao, K., and Zalesky, A. (2020). Structure-function coupling in the human connectome: A machine learning approach. *NeuroImage*, page 117609.
- Sinclair, B., Hansell, N. K., Blokland, G. A., Martin, N. G., Thompson, P. M., Breakspear, M., de Zubicaray, G. I., Wright, M. J., and McMahon, K. L. (2015). Heritability of the Network Architecture of Intrinsic Brain Functional Connectivity. *NeuroImage*, 121:243.
- Smith, R. E., Tournier, J.-D., Calamante, F., and Connelly, A. (2015). Sift2: Enabling dense quantitative assessment of brain white matter connectivity using streamlines tractography. *Neuroimage*, 119:338–351.
- Suárez, L. E., Markello, R. D., Betzel, R. F., and Misic, B. (2020). Linking structure and function in macroscale brain networks. *Trends in Cognitive Sciences*.
- Torregrossa, M. M., Quinn, J. J., and Taylor, J. R. (2008). Impulsivity, compulsivity, and habit: the role of orbitofrontal cortex revisited. *Biological psychiatry*, 63(3):253–255.
- Vuoksima, E., Panizzon, M. S., Hagler, D. J., Hatton, S. N., Fennema-Notestine, C., Rinker, D., Eyler, L. T., Franz, C. E., Lyons, M. J., Neale, M. C., Tsuang, M. T., Dale, A. M., and Kremen, W. S. (2017). Heritability of white matter microstructure in late middle age: A twin study of tract-based fractional anisotropy and absolute diffusivity indices. *Human Brain Mapping*, 38(4):2026–2036.
- Whitfield-Gabrieli, S. and Nieto-Castanon, A. (2012). Conn: a functional connectivity toolbox for correlated and anticorrelated brain networks. *Brain connectivity*, 2(3):125–41.
- Yeo, B. T., Krienen, F. M., Sepulcre, J., Sabuncu, M. R., Lashkari, D., Hollinshead, M., Roffman, J. L., Smoller, J. W., Zöllei, L., Polimeni, J. R., et al. (2011). The organization of the human cerebral cortex estimated by intrinsic functional connectivity. *Journal of neurophysiology*.
- Zhao, B., Zhang, J., Ibrahim, J. G., Luo, T., Santelli, R. C., Li, Y., Li, T., Shan, Y., Zhu, Z., Zhou, F., Liao, H., Nichols, T. E., and Zhu, H. (2019). Large-scale GWAS reveals genetic architecture of brain white matter microstructure and genetic overlap with cognitive and mental health traits (n = 17,706). *Molecular Psychiatry*, pages 1–13.

REVIEWER COMMENTS

Reviewer #2 (Remarks to the Author):

I appreciate the extremely thorough revision that has addressed nearly all of my comments. The authors should be congratulated on their excellent work.

I have only one remaining comment, regarding the comparison of regional profiles. As prior, I again suggest use of the spin test or brainsmash. A simple permutation test where spatial structure is not preserved is an extremely weak null that (those prior papers show) does not adequately control the type I error rate. Assessing significance of correspondence between maps requires accounting for spatial autocorrelation.

Reviewer #3 (Remarks to the Author):

I thank the authors for the considerable revisions. However, my concerns remain and have been amplified by the current outcome. Please find my reservations described in more detail below:

1. I fail to understand whether a correlation of two maps warrants the conclusion that heritability of SC-FC coupling is not 'driven' by FC and SC heritability. Conceptually, what would this mean? Heritability is the amount of variance that can be accounted for by genetic effects whereas the SC-FC coupling relates to the correlation of FC and SC. Would it not be more conceptually clear to correct for FC/SC of the involved nodes in the model when computing heritability of SC-FC coupling? And even when doing this, it cannot be ruled out that absence of evidence does not mean it does not exist.
2. The statement of figure 2 stating that SC-FC varies across the brain and is related by within and between network coupling? Isn't the latter statement circular given that the SC-FC coupling was based on both within and between network coupling? So that it would be strange if there would be no relation at all?
3. In the response to reviewer 1 it is stated that the SC-FC coupling is driven by between-network correlations, but looking at this map many regions show a very low association <0.1 so I am unsure how to interpret this? How can this then explain the pattern beyond showing that there is a correlation between both maps?
4. The spatial autocorrelation comment by R2 is not correctly addressed to my understanding, e.g. if a correlation is beyond 'random' spatial autocorrelations between maps. Performing a permutation does not answer the question.
5. I am not confident that the weak association observed with age in the current paper would be sufficient evidence to highlight it in the title. Though the authors state that this region are in line with previous reports of Baum et al, this is only in part correct and highest links are not found in this region.
6. Would the sex difference have any relationship with difference in brain size in males and females?
7. At the end of the abstract it was stated that SC_FC coupling was highly heritable. I believe the mean score is about 0.4? Would this be considered highly heritable?
8. There are various typo's in the manuscript (for example 'pareital' on page 7).

Minor comment:

a. Apologies for bringing this up now, but I am unsure whether the use of jet-colormap is best to display SC-FC coupling. Why was this colormap chosen? This biases the visualisation making difference between values seem both higher and lower depending on the range of the colormap.

Reviewer #4 (Remarks to the Author):

The authors made a tremendous effort to address the points raised by reviewers and I believe that the final outcome is a nice, strong paper.

Feedback from Reviewer #2 on Reviewer #3's concerns:

>1. I fail to understand whether a correlation of two maps warrants the conclusion that heritability of SC-FC coupling is not 'driven' by FC and SC >heritability. Conceptually, what would this mean? Heritability is the amount of variance than can be accounted for by genetic effects whereas the >SC-FC coupling relates to the correlation of FC and SC. Would it not be more conceptually clear to correct for FC/SC of the involved nodes in the >model when computing heritability of SC-FC coupling? And even when doing this, it cannot be ruled out that absence of evidence does not mean it >does not exist.

Response: It is not unreasonable for the reviewer to ask the authors to show heritability of SC/FC "above and beyond" the component parts. The authors could do this by conditioning on either structural or functional connectivity in their analysis.

>2. The statement of figure 2 stating that SC-FC varies across the brain and is related by within and between network coupling? Isn't the latter >statement circular given that the SC-FC coupling was based on both within and between network coupling? So that it would be strange if there >?>would be no relation at all?

Response: More than anything, the analyses of fig 2, panels d, e, and f could be better motivated and described. On re-reading now, it is somewhat unclear what the authors are attempting to learn here, and how to interpret the results given that the within/between network SC-Fc is a component of whole brain SC-FC relationship (as noted by the reviewer).

3. In the response to reviewer 1 it is stated that the SC-FC coupling is driven by between-network correlations, but looking at this map many regions show a very low association <0.1 so I am unsure how to interpret this? How can this then explain the pattern beyond showing that there is a correlation between both maps?

Response: I do not in fact follow this comment (to be perfectly honest).

4. The spatial autocorrelation comment by R2 is not correctly addressed to my understanding, e.g. if a correlation is beyond 'random' spatial autocorrelations between maps. Performing a permutation does not answer the question.

Response: I agree (as noted in my review).

5. I am not confident that the weak association observed with age in the current paper would be sufficient evidence to highlight it in the title. Though the authors state that this region are in line with previous reports of Baum et al, this is only in part correct and highest links are not found in this region.

Response: The one region related to age survived appropriate multiple comparison correction; it does in fact overlap with the previous report from Baum et al.

6. Would the sex different have any relationship with difference in brain size in males and females?

Response: This should be accounted for by covarying for ICV; the authors did that in their model.

7. At the end of the abstract it was stated that SC_FC coupling was highly heritable. I believe the mean score is about 0.4? Would this be considered highly heritable?

Response: May be worth tempering as noted by the reviewer.

Response To Reviewer #2

Overall Comments

I appreciate the extremely thorough revision that has addressed nearly all of my comments. The authors should be congratulated on their excellent work.

I have only one remaining comment, regarding the comparison of regional profiles. As prior, I again suggest use of the spin test or brainsmash. A simple permutation test where spatial structure is not preserved is an extremely weak null that (those prior papers show) does not adequately control the type I error rate. Assessing significance of correspondence between maps requires accounting for spatial autocorrelation.

Response

Thank you for your recognition of our revised manuscript. We apologize for not fully understanding your previous comment on the regional profiles comparison. We have reanalyzed the network comparisons and correlations using BrainSMASH and updated the results and figures. The results are largely the same, with some slight differences in significance between a few of the network comparisons. We thank you for your comments that have greatly strengthened the manuscript.

Response To Reviewer #3

Overall Comments

I thank the authors for the considerable revisions. However, my concerns remain and have been amplified by the current outcome. Please find my reservations described in more detail below.

Response

We would like to thank you for your feedback. We believe the changes to our analyses and explanations added in response to your comments has strengthened the manuscript.

Reviewer 3's Comment

I fail to understand whether a correlation of two maps warrants the conclusion that heritability of SC-FC coupling is not 'driven' by FC and SC heritability. Conceptually, what would this mean? Heritability is the amount of variance than can be accounted for by genetic effects whereas the SC-FC coupling relates to the correlation of FC and SC. Would it not be more conceptually clear to correct for FC/SC of the involved nodes in the model when computing heritability of SC-FC coupling? And even when doing this, it cannot be ruled out that absence of evidence does not mean it does not exist.

Reviewer 2's Response To Reviewer 3's Comment

It is not unreasonable for the reviewer to ask the authors to show heritability of SC/FC "above and beyond" the component parts. The authors could do this by conditioning on either structural or functional connectivity in their analysis.

Response

Thank you for your comments. We agree that it would be beneficial to account for SC/FC of the regions when calculating the heritability of SC-FC coupling. Thus, we re-calculated the heritability of SC-FC coupling after including SC degree and FC degree as regional fixed effect covariates in the model. Our updated results and figure in the manuscript are shown below in Figure 1. We also note that including SC degree and FC degree as covariates does not change the heritability estimates for SC-FC coupling, see Figure 2. Finally, we agree that the term 'driven' may be used inappropriately in regards to the findings, thus we have changed the phrasing to either say "SC-FC coupling is heritable and different from FC or SC heritability" or "the heritability of SC-FC coupling is not strongly associated with FC or SC heritability". We have also added 3 Supplemental Figures (13-15), showing the variance explained by each of the model components in the heritability estimates for completeness.

Furthermore, in making the above change in the heritability model, we discovered an error in our prior heritability calculation - in the last version of the manuscript, we were reporting the total fraction of variance explained by genetics AND common/individual environment out of

Figure 1: Heritability of SC-FC coupling (accounting for FC and SC as fixed-effects covariates in the model), and FC and SC node strength, respectively.

Figure 2: Heritability of SC-FC coupling before including SC and FC node strength as covariates, and the relationship between the uncorrected and corrected heritability values.

the total explained by genetics, common/individual environment and measurement error, $(\sigma_A^2 + \sigma_C^2 + \sigma_E^2)/(\sigma_A^2 + \sigma_C^2 + \sigma_E^2 + \sigma_M^2)$, instead of the total fraction of stable, non-transient intersubject variance explained by just the genetic variance $\sigma_A^2/(\sigma_A^2 + \sigma_C^2 + \sigma_E^2)$ as it should have been. We apologize for this error, which resulted in an underestimate of the actual heritability values for FC node strength and SC-FC coupling (see changes in abstract, results and discussion sections). Most importantly, this is the new paragraph in the discussion:

"For the first time, we show that regional SC-FC coupling is highly heritable across the brain, particularly in subcortical, cerebellar/brainstem and visual networks. Measurement noise in subcortical regions is highest among the networks, which may suggest increased uncertainty in those regions' heritability estimates (see Supplemental Fig. 13). We find that regional SC-FC coupling heritability is of similar magnitude to FC heritability, and that both are more heritable than SC. Furthermore, we saw that SC-FC coupling heritability was not substantially explained by either SC or FC node strength heritability; in fact, it was only moderately correlated with FC node strength heritability and not correlated with SC node strength heritability. Previous studies have shown heritability of FC profiles, with the default mode network having highest heritability (estimates ranging from 0.42–0.8) Ge et al. (2017); Glahn et al. (2010). Our results showed heritability of FC degree in default mode network was indeed significantly higher than other higher-order cortical networks, but not significantly different from visual or somatomotor networks and significantly lower than limbic, subcortical and cerebellar/brainstem networks. Some discrepancy with earlier work may arise from the fact we were measuring heritability of node degree rather than pairwise connections as well as differences in the model used to estimate heritability. Limbic regions in particular had highest heritability among the cortical networks for FC node strength, which contradicts some previous work. However, we observe that the total amount of variance explained by genetics and common/individual environment were lowest and the standard error of the fraction of total variance explained by genetics and common/individual environment were the highest in the limbic network (see Supplemental Fig. 14), indicating possible increased uncertainty in those regions' heritability estimates. From the reliability analysis, it does not appear that the SC's lower heritability values are due to increased measurement noise, as SC node strength was as reliable as FC and SC-FC coupling. Note, however, that since we only have one SC measurement per subject, our approach can not account for within-subject measurement error when estimating the heritability of SC, which might explain some of the differences compared to FC and SC-FC coupling. Previous work has suggested different genetic signatures underlying brain anatomy and physiology Glahn et al. (2010); here, heritability of the two modalities' node strengths were indeed not correlated. One recent study quantifying anatomical heritability of the size of cortical areas (as defined by FC) showed unimodal motor/sensory networks had higher heritability (0.44) relative to heteromodal association networks (0.33) Anderson et al. (2021). We do observe partial agreement with their findings in that unimodal visual networks, but not somato-motor networks, had higher anatomical SC heritability compared to many other cortical networks."

Reviewer 3's Comment

The statement of figure 2 stating that SC-FC varies across the brain and is related by within and between network coupling? Isn't the latter statement circular given that the SC-FC coupling was based on both within and between network coupling? So that it would be strange if there would be no relation at all?

Reviewer 2's Response

More than anything, the analyses of fig 2, panels d, e, and f could be better motivated and described. On re-reading now, it is somewhat unclear what the authors are attempting to learn here, and how to interpret the results given that the within/between network SC-FC is a component of whole brain SC-FC relationship (as noted by the reviewer).

Our Response

It is true that the between and within-network SC-FC coupling are components of whole-brain SC-FC coupling; the point of the analysis was to compare the associations of between and within network coupling with whole brain coupling to determine which one was more related to whole-brain coupling. We have added some additional motivation in the Results section (see text below). Of course, due to the difference in sheer number of regions in the within and between network analysis, the whole brain coupling is more similar to the between network coupling. We acknowledge that this analysis is not central to the paper, so we have moved the figures to the supplementary document (Figure S2) to de-emphasize their (rather obvious) conclusion in addition to removing some discussion of the results in the text (see below for revised text in the Results and Discussion sections).

"This whole-brain measure of SC-FC coupling reflects the alignment of a region's functional and structural connectivity profiles to every other region in the brain, but it does not disentangle the contribution of between or within network connections to the whole-brain coupling value. To assess the association between whole-brain SC-FC coupling and between and within-network coupling, we separately calculated, for each region, its between and within-network SC-FC coupling. Within-network SC-FC coupling for each region was the Spearman correlation of the structural and functional connections between that region and other regions in the same network; between-network SC-FC coupling the same calculation but between that region and regions outside of its assigned network."

"When comparing whole-brain SC-FC coupling to the within and between-network coupling, we found that, unsurprisingly, whole brain coupling was highly correlated with the between-network SC-FC coupling (Pearson's $r = 0.704$, $p = 0$) and moderately correlated with the within-network coupling (Pearson's $r = 0.416$, $p = 0$), see Supplementary Information Figure S2. This is likely due to the much larger number of between-network region-pairs than within-network region-pairs in the whole-brain SC-FC coupling calculations."

Reviewer 3's Comment

In the response to reviewer 1 it is stated that the SC-FC coupling is driven by between-network correlations, but looking at this map many regions show a very low association < 0.1 so I am unsure how to interpret this? How can this then explain the pattern beyond showing that there is a correlation between both maps?

Reviewer 2's Response

I do not in fact follow this comment (to be perfectly honest).

Our Response

Unfortunately, we do not quite understand reviewer 2's comment either - but we do hope that our changes to the manuscript de-emphasizing the between and within-network analysis have satisfied the reviewer's concerns.

Reviewer 3's Comment

The spatial autocorrelation comment by R2 is not correctly addressed to my understanding, e.g. if a correlation is beyond 'random' spatial autocorrelations between maps. Performing a permutation does not answer the question.

Reviewer 2's Response

I agree (as noted in my review).

Our Response

Thank you for addressing this. We have changed our permutation test to using BrainSMASH (Burt et al., 2020) to account for the spatial autocorrelations, and updated the results and figures in the manuscript and supplementary material (which are largely unchanged).

Reviewer 3's Comment

I am not confident that the weak association observed with age in the current paper would be sufficient evidence to highlight it in the title. Though the authors state that this region are in line with previous reports of Baum et al, this is only in part correct and highest links are not found in this region.

Reviewer 2's Response

The one region related to age survived appropriate multiple comparison correction; it does in fact overlap with the previous report from Baum et al.

Our Response

We actually did observe several regions that showed significant associations (after FDR correction) with age so we think it is sufficient for inclusion in the title. Our significant age-related increases were all in default mode network which is indeed in agreement with Baum et al. (replication in different datasets across varying ages is important), and our age-related decreases were mostly distributed in cerebellum/brain stem which the previous work didn't contain (thus it is a novel finding that should be highlighted). Thus we think the results regarding age are justified for inclusion in the title.

Reviewer 3's Comment

Would the sex different have any relationship with difference in brain size in males and females?

Reviewer 2's Response

This should be accounted for by covarying for ICV; the authors did that in their model.

Our Response

Thank you for this question. We did in fact account for ICV in our GLM, so any sex differences would be accounted for in the model.

Reviewer 3's Comment

At the end of the abstract it was stated that SC-FC coupling was highly heritable. I believe the mean score is about 0.4? Would this be considered highly heritable?

Reviewer 2's Response

May be worth tempering as noted by the reviewer.

Response

Thank you for this comment. As mentioned, during the revision we identified an error in the heritability calculations that had resulted in an underestimate of true heritability levels in the previous version of the manuscript. After correcting the error and additionally adding in SC and FC node strength as covariates in the heritability model, we found SC-FC coupling to indeed have high heritability (0.5-0.9) therefore we chose not to change our original phrasing.

Reviewer Comment

There are various typo's in the manuscript (for example 'pareital' on page 7).

Response

Thank you for pointing out this error. We have carefully examined the manuscript to correct any outstanding typos.

Reviewer Comment

Apologies for bringing this up now, but I am unsure whether the use of jet-colormap is best to display SC-FC coupling. Why was this colormap chosen? This biases the visualisation making difference between values seem both higher and lower depending on the range of the colormap.

Response

Thank you for this comment. The jet colormap was chosen as it contains a wide range of colors. After considering your suggestion, we changed all the brain visualizations that used jet to instead use magma, which is more continuous from dark to light across the range of values.

Response To Reviewer #4

Overall Comments

The authors made a tremendous effort to address the points raised by reviewers and I believe that the final outcome is a nice, strong paper.

Response

We would like to thank you for your recognition of the revised manuscript.

References

- Anderson, K. M., Ge, T., Kong, R., Patrick, L. M., Spreng, R. N., Sabuncu, M. R., Yeo, B. T., and Holmes, A. J. (2021). Heritability of individualized cortical network topography. *Proceedings of the National Academy of Sciences*, 118(9).
- Burt, J. B., Helmer, M., Shinn, M., Anticevic, A., and Murray, J. D. (2020). Generative modeling of brain maps with spatial autocorrelation. *NeuroImage*, 220:117038.
- Ge, T., Holmes, A. J., Buckner, R. L., Smoller, J. W., and Sabuncu, M. R. (2017). Heritability analysis with repeat measurements and its application to resting-state functional connectivity. *Proceedings of the National Academy of Sciences*, 114(21):5521–5526.
- Glahn, D. C., Winkler, A. M., Kochunov, P., Almasy, L., Duggirala, R., Carless, M. A., Curran, J. C., Olvera, R. L., Laird, A. R., Smith, S. M., Beckmann, C. F., Fox, P. T., and Blangero, J. (2010). Genetic control over the resting brain. *Proceedings of the National Academy of Sciences*, 107(3):1223–1228.

REVIEWERS' COMMENTS

Reviewer #2 (Remarks to the Author):

No further comments; all my concerns have been adressed!

Reviewer #3 (Remarks to the Author):

I thank the authors for their hard work in replying to my comments and updating their manuscript.

Response To Reviewer #2

Overall Comments

No further comments; all my concerns have been addressed!

Response

Thank you so much for all the constructive comments and suggestions. We are glad that our revision addressed all your concerns.

Response To Reviewer #3

Overall Comments

I thank the authors for their hard work in replying to my comments and updating their manuscript.

Response

Thank you so much for providing many constructive feedback and recognizing our work. We are very glad that our revision satisfy your requirement.